# Bridging the Gap: Providing Post-Hoc Symbolic Explanations for Sequential Decision-Making Problems with Inscrutable Representations

## Abstract

As increasingly complex AI systems are introduced into our daily lives, it becomes important for such systems to be capable of explaining the rationale for their decisions and allowing users to contest these decisions. A significant hurdle to allowing for such explanatory dialogue could be the *vocabulary mismatch* between the user and the AI system. This paper introduces methods for providing contrastive explanations in terms of user-specified concepts for sequential decision-making settings where the system's model of the task may be best represented as a blackbox simulator. We do this by building partial symbolic models of a local approximation of the task that can be leveraged to answer the user queries. We empirically test these methods on a popular Atari game (Montezuma's Revenge) and modified versions of Sokoban (a well known planning benchmark) and report the results of user studies to evaluate whether people find explanations generated in this form useful.

## 1 Introduction

For AI systems to be truly effective in the real world, they need to not only be capable of coming up with near-optimal decisions but also be capable of working effectively with their end users. One of the key requirements for such collaboration would be to allow users to raise explanatory queries wherein they can contest the system's decisions. An obstacle to providing explanations to such questions is the fact that the systems may not have a shared vocabulary with its end users or have an explicit interpretable model of the task. More often than not, the system may be reasoning about the task in a high-dimensional space that is opaque to even the developers of the system, let alone a lay user.

While there is a growing consensus within the explainable AI community that end-user explanations need to be framed in terms of user understandable concepts, the focus generally has been on introducing such methods for explaining one-shot decisions such as in the case of classifiers (c.f. Kim et al. (2018); Ribeiro et al. (2016)). This is unfortunate as explaining sequential decision-making problems presents many challenges that are absent from the one-shot decision-making scenarios. In such problems, we not only have to deal with possible interrelationship between the actions in the sequence, but may also need to explain conditions for the executability of actions and the cost of executing certain action sequences. Effectively, this means that explaining a plan or policy to a user would require the system to explain the details of the domain (or at least the agent's belief of it).

In this paper, we propose methods that are able to field some of the most fundamental explanatory queries identified in the literature, namely contrastive queries, i.e., questions of the form 'why P (the decision proposed by the system) and not Q (the alternative proposed by the user or the foil)?' (Miller, 2018), in user-understandable terms. Our methods achieve this by building partial and abstract symbolic models (Section 2) expressed in terms of the user's vocabulary that approximates task details relevant to the specific query raised by the user. To the best of our knowledge, we are the first work to propose learning of symbolic local approximations of the problem dynamics and cost function for explanations in sequential decision-making scenarios. Specifically, we will focus on

deterministic tasks where the system has access to a task simulator and we will identify (a) missing preconditions to explain scenarios where the foil raised by the user results in an execution failure of action and (b) cost function approximations to explain cases where the foil is executable but suboptimal (Section 3). We *learn* such models by interacting with the simulator (on randomly sampled states) while using learned classifiers that detect the presence of user-specified concepts in the simulator states. Figure 1 presents the overall flow of this process with illustrative explanations in the context of a slightly updated version of Montezuma's Revenge (Wikipedia contributors, 2019). Our methods also allow for the calculation of confidence over the explanations and explicitly take into account the fact that learned classifiers for user-specified concepts may be noisy. This ability to quantify its belief about the correctness of explanation is an important capability for any post-hoc explanation system that may influence the user's decisions. We evaluate the system on two popular sequential decision-making domains, Montezuma's Revenge and modified versions of Sokoban (Botea et al., 2002) (a game involving players pushing boxes to specified targets). We present user study results that show the effectiveness of explanations studied in this paper (Section 5).

## 2 BACKGROUND

Our focus here is to address cases where a user is trying to make sense of agent behavior that may in fact be optimal in the agent's model of the task but is confusing to the user owing to a differing understanding of the task (or them overlooking some facts about the task).

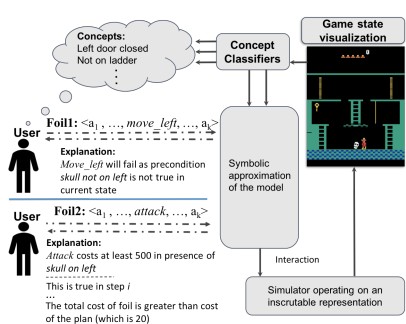

Thus our focus isn't on how the agent came up with the specific decisions, but only on why this action sequence was chosen instead of an alternative that the user expected. We assume agent has access to a deterministic simulator of the form $\mathcal{M}_{\text{sim}} = \langle S, A, T, \mathcal{C} \rangle$, where $S$ represents the set of possible world states, $A$ the set of actions and $T$ the transition function that specifies the task dynamics. The transition function is defined as $T : S \times A \to S \cup \{\bot\}$, where $\bot$ corresponds to an invalid absorber-state generated by the execution of an infeasible action. Invalid state could be used to capture failure states that could occur when the agent violates hard constraints like safety constraints. Finally, $\mathcal{C} : S \times A \to \mathbb{R}$ captures the cost function of executing an action at a particular state (with the cost of an infeasible action taken to be infinite). We will overload the transition function and cost functions to also take in a sequence of actions (which in the case of the transition function returns the final state resulting from executing the action sequence and for cost function the cost of executing the sequence actions). We will consider goal-directed agents that are trying to drive the state of the world to one of the goal states. Where the tuple $\Pi_{\text{sim}} = \langle I, \mathbb{G}, \mathcal{M}_{\text{sim}} \rangle$ represents the agent's decision making problem ($I$ is the initial state and $\mathbb{G}$ the set of goal states). The agent comes up with a plan (a sequence of actions) $\pi$ such that $T(I, \pi) \in \mathbb{G}$ and

Figure 1: *Explanatory dialogue starts when the user presenting a specific alternate plan (foil). We consider two foils, one that is invalid and another that is costlier than the plan. System explains the invalid plan by pointing out an action precondition that was not met in the plan, while it explains the foil suboptimality by informing the user about cost function. These model information are expressed in terms of concepts specified by the user which we operationalize by learning classifiers for each concept.*

the plan is said to be optimal if there exists no cheaper plan that achieves the goal.

We will use symbolic action models with preconditions and cost functions (similar to STRIPS models (Geffner & Bonet, 2013)) as a way to approximate the problem for explanations. Such a model can be represented by the tuple $\mathcal{M}_{\mathcal{S}} = \langle F_{\mathcal{S}}, A_{\mathcal{S}}, I_{\mathcal{S}}, G_{\mathcal{S}}, \mathcal{C}_{\mathcal{S}} \rangle$, where $F_{\mathcal{S}}$ is a set of propositional fluents defining the state space, $A_{\mathcal{S}}$ is the set of actions, $I_{\mathcal{S}}$ is the initial state, $G_{\mathcal{S}}$ is the goal specification. Each valid problem state in the problem is uniquely identified by the subset of fluents that are true in that state (so for any state $s \in S_{\mathcal{M}_{\mathcal{S}}}$, where $S_{\mathcal{M}_{\mathcal{S}}}$ is the set of states for $\mathcal{M}_{\mathcal{S}}$, $s \subseteq F_{\mathcal{S}}$). Each action $a \in A_{\mathcal{S}}$ is further described in terms of the preconditions $prec_a$ (specification of states in which $a$ is executable) and the effects of executing the action. We will denote the state formed by executing action $a$ in a state $s$ as $a(s)$. We will focus on models where the preconditions are represented as a conjunction of state factors. If the action is executed in a state with missing preconditions, then the execution results in the invalid state ($\bot$). Unlike standard

STRIPS models, where the cost of executing action is independent of states, we will be using a state dependent cost function of the form $\mathcal{C}_S : 2^F \times A_S \to \mathbb{R}$ to capture the cost of valid action executions. Internally, such state models may be represented using conditional cost models of the type discussed in (Geißer et al., 2016). In this paper, we won't try to reconstruct the exact cost function but will rather try to estimate an abstract version of the cost function.

## 3    Contrastive Explanations

The specific explanatory setting, illustrated in Figure 1, that we are interested in studying involves a decision-making problem specified by the tuple $\Pi_{sim} = \langle I, \mathbb{G}, \mathcal{M}_{\text{sim}} \rangle$ for which the system identifies a plan $\pi$. When presented with the plan, the user of the system may either accept it or responds by raising an alternative plan $\pi_f$ (*the foil*) that they believe should be followed instead. Now the system would need to provide an explanation as to why the plan $\pi$ may be preferred over the foil $\pi_f$, by showing that the foil is invalid or it is costlier than the plan.[1] More formally,

**Definition 1** *The plan $\pi$ is said to be preferred over a foil $\pi_f$ for a problem $\Pi_{sim} = \langle I, \mathbb{G}, \mathcal{M}_{sim} \rangle$, if either of the following conditions are met, i.e., (1) $\pi_f$ is invalid, if (a) $T(I, \pi_f) \notin \mathbb{G}$, i.e the action sequence doesn't lead to a possible goal state, or (b) the execution of the plan leads to an invalid state, i.e., $T(I, \pi_f) = \bot$. (2) Or $\pi_f$ is* costlier *than $\pi$, i.e., $\mathcal{C}(I, \pi) < \mathcal{C}(I, \pi_f)$*

To concretize this interaction, consider the modified version of Montezuma's revenge (Figure 1). The agent starts from the highest platform, and the goal is to get to the key. The specified plan $\pi$ may require the agent to make its way to the lowest level, jump over the skull, and then go to the key with a total cost of 20. Now the user raises two possible foils that are quite similar to $\pi$, but, (a) in the first foil, instead of jumping the agent just moves left (as in it tries to move through the skull) and (b) in the second, instead of jumping over the skull, the agent performs the attack action (not part of the original game, but added here for illustrative purposes) and then moves on to the key. Using the simulator, the system could tell that in the first case, moving left would lead to an invalid state, and in the second case, the foil is costlier. Though it may struggle to explain what particular aspects of the state or state sequence lead to the invalidity or suboptimality. Even if it could identify parts of its own internal state representation as an explanation, it would not necessarily be meaningful to the end-user. This scenario thus necessitates the use of methods that are able to express possible explanations in terms that the user may understand.

**Concept maps:** In this setting, the system would need a mapping from its internal state representation to a set of high-level concepts that are known to the user (for Montezuma this could involve concepts like *agent being on a ladder, holding onto the key, being next to the skull, etc.*). We will assume each concept corresponds to a propositional fact that the user associates with the task's states and believes that the dynamics of the task are determined by these concepts (the specifics of the representational assumptions made by this paper can be found in the appendix). This means that as per the user, for each given state, a subset of these concepts may be either present or absent. We will assume access to binary classifiers for each of these concepts. These classifiers provide us with a way to convert simulator states to a factored representation. Such techniques have not only been used in explanation (c.f. Kim et al. (2018); Hayes & Shah (2017)) but also in works that have looked at learning high-level representations for continuous state-space problems (c.f. Konidaris et al. (2018)). Let $\mathbb{C}$ be the set of classifiers corresponding to the high-level concepts. For state $s \in S$, we will overload the notation $\mathbb{C}$ and specify the concepts that are true as $\mathbb{C}(s)$, i.e., $\mathbb{C}(s) = \{c_i | c_i \in \mathbb{C} \wedge c_i(s) = 1\}$ (where $c_i$ is the classifier corresponding to the $i^{th}$ concept and we overload the notation to also stand for the label of the $i^{th}$ concept). The user could specify the set of concepts by identifying positive and negative example states for each concept. These examples could then be used to learn the required classifiers by using algorithms best suited for the internal simulator state representation. Thus the system should have some method of exposing simulator states to the user. A common way to satisfy this requirement would be by having access to visual representations for the states. The simulator state itself doesn't need to be an image as long as we have a way to visualize it (for example in Atari games where the states can be represented by the RAM state of the game controller but we can still visualize them).

**Explanation using concepts:** To explain why a given foil is not preferred over the specified plan, we will present information about the symbolic model expressed in user's vocabulary, $\mathcal{M}_S^{\mathbb{C}} =$

---

[1]If the foil is as good as the original plan, then the system could switch to foil without loss of optimality.

$\langle \mathbb{C}, A_{\mathcal{S}}^{\mathbb{C}}, \mathbb{C}(I), \mathbb{C}(\mathbb{G}), \mathcal{C}_{\mathcal{S}}^{\mathbb{C}} \rangle$. Where $\mathbb{C}(\mathbb{G}) = \bigcap_{s_g \in \mathbb{G}} \mathbb{C}(s_g)$ and $A_{\mathcal{S}}^{\mathbb{C}}$ contains a definition for each action $a \in A$. The model is said to be a *local symbolic approximation* for the simulator for regions of interest $\hat{S} \subseteq S$ if $\mathcal{M}_{\mathcal{S}}^{\mathbb{C}}$ is a sound abstraction of the simulator $\mathcal{M}_{\text{sim}} = \langle S, A, T, \mathcal{C} \rangle$ for $\hat{S}$. That is $\forall s \in \hat{S}$ and $\forall a \in A$, we have an equivalent action $a^{\mathbb{C}} \in A_{\mathcal{S}}^{\mathbb{C}}$, such that $a^{\mathbb{C}}(\mathbb{C}(s)) = \mathbb{C}(T(s,a))$ (assuming $\mathbb{C}(\bot) = \bot$) and $\mathcal{C}_{\mathcal{S}}^{\mathbb{C}}(\mathbb{C}(s), a) = \mathcal{C}(s,a)$. Note that establishing the preference of plan does not require informing the users about the entire model, but rather only the relevant parts. For conciseness, we will use $a_i$ for both the simulator action and the corresponding abstract action in the symbolic model as long as the context allows it to be distinguished.

To establish invalidity of $\pi_f$, we just need to focus on explaining the failure of the first failing action $a_i$, i.e., the last action in the shortest prefix that would lead to an invalid state (move-left action in the state presented in Figure 1 for the first foil). We can do so by informing the user that the action has an unmet precondition, as per the symbolic model, in the failing state. Formally

**Definition 2** *For a failing action $a_i$ for the foil $\pi_f = \langle a_1, .., a_i, .., a_n \rangle$, $c_i \in \mathbb{C}$ is considered an explanation for failure if $c_i \in prec_{a_i} \setminus \mathbb{C}(s_i)$, where $s_i$ is the state where $a_i$ is meant to be executed (i.e $s_i = T(I, \langle a_1, .., a_{i-1} \rangle)$).*

In the example invalid foil, the explanation would inform the user that move-left can only be executed in states for which the concept skull-not-on-left is true; and the concept is false in the given state. This formulation is enough to capture both conditions for foil invalidity by appending an additional goal action at the end of each sequence. The goal action causes the state to transition to an end state and it fails for all states except the ones in $\mathbb{G}$. Our approach to identifying the minimal explanation for specific query follows from studies in social sciences that have shown that selectivity or minimality is an essential property of effective explanations Miller (2018).

For explaining suboptimality, we have to inform the user about $\mathcal{C}_{\mathcal{S}}^{\mathbb{C}}$. To ensure minimality of explanations, rather than provide the entire cost function or even the individual conditional components of the function, we will instead try to learn and provide an abstraction of the cost function $\mathcal{C}_s^{abs}$

**Definition 3** *For the symbolic model $\mathcal{M}_{\mathcal{S}}^{\mathbb{C}} = \langle \mathbb{C}, A_{\mathcal{S}}^{\mathbb{C}}, \mathbb{C}(I), \mathbb{C}(\mathbb{G}), \mathcal{C}_{\mathcal{S}}^{\mathbb{C}} \rangle$, an abstract cost function $\mathcal{C}_{\mathcal{S}}^{abs} : 2^{\mathbb{C}} \times A_{\mathcal{S}}^{\mathbb{C}} \to \mathbb{R}$ is specified as follows $\mathcal{C}_{\mathcal{S}}^{abs}(\{c_1, .., c_k\}, a) = min\{\mathcal{C}_{\mathcal{S}}^{\mathbb{C}}(s, a) | s \in S_{\mathcal{M}_{\mathcal{S}}^{\mathbb{C}}} \wedge \{c_1, .., c_k\} \subseteq s\}$.*

Intuitively, $\mathcal{C}_{\mathcal{S}}^{abs}(\{c_1, .., c_k\}, a) = k$ can be understood as stating that *executing the action $a$, in the presence of concepts $\{c_1, .., c_k\}$ costs at least $k$.* We can use $\mathcal{C}_{\mathcal{S}}^{abs}$ in an explanation of the form

**Definition 4** *For a valid foil $\pi_f = \langle a_1, .., a_k \rangle$, a plan $\pi$ and a problem $\Pi_{sim}$, the sequence of concept sets of the form $\mathbb{C}_{\pi_f} = \langle \hat{\mathbb{C}}_1, ..., \hat{\mathbb{C}}_k \rangle$ along with $\mathcal{C}_s^{abs}$ is considered a valid explanation for relative suboptimality of the foil (denoted as $\mathcal{C}_{\mathcal{S}}^{abs}(\mathbb{C}_{\pi_f}, \pi_f) > \mathbb{C}(I, \pi)$), if $\forall \hat{\mathbb{C}}_i \in \mathbb{C}_{\pi_f}$, $\hat{\mathbb{C}}_i$ is a subset of concepts presents in the corresponding state (where state is $I$ for $i = 1$ and $T(I, \langle a_1, ..., a_{i-1} \rangle)$ for $i > 1$) and $\Sigma_{i = \{1..k\}} \mathcal{C}_{\mathcal{S}}^{abs}(\hat{\mathbb{C}}_i, a_i) > \mathcal{C}(I, \pi)$*

In the earlier example, the explanation would include the fact that executing the action attack in the presence of the concept skull-on-left, will cost at least 500 (as opposed to original plan cost of 20).

## 4 IDENTIFYING EXPLANATIONS THROUGH SAMPLE-BASED TRIALS

For identifying the model parts for explanatory query, we will rely on the agent's ability to interact with the simulator to build estimates. Given the fact that we can separate the two cases at the simulator level, we will keep the discussion of identifying each explanation type separate and only focus on identifying the model parts once we know the failure type.

**Identifying failing precondition:** To identify the missing preconditions, we will rely on the simple intuition that while successful execution of an action $a$ in the state $s_j$ with a concept $C_i$ doesn't necessarily establish that $C_i$ is a precondition, we can guarantee that any concept false in that state can not be a precondition of that action. This is a common line of reasoning exploited by many of the model learning methods (c.f Carbonell & Gil (1990); Stern & Juba (2017)). We start with the set of concepts that are absent in the the state ($s_{\text{fail}}$) where the failing action ($a_{\text{fail}}$) was executed, i.e., poss_prec_set $= \mathbb{C} \setminus \mathbb{C}(s_{\text{fail}})$. We then randomly sample for states where $a_{\text{fail}}$ is executable. Each new sampled state $s_i$ where the action is executable can then be used to update the possible precondition set as poss_prec_set $=$ poss_prec_set $\cap \mathbb{C}(s_i)$. That is, if a state is identified where the action is executable but a concept is absent then it can't be part of the precondition. We will keep repeating this sampling step until the sampling budget is exhausted or if one of the following exit conditions is met. (a) In cases where we are guaranteed that the concept list is exhaustive, we can quit as soon

as the set of possibilities reduce to one (since there has to be a missing precondition at the failure state). (b) The search results in an empty list. The list of concepts left at the end of exhausting the sampling budget represents the most likely candidates for preconditions. *An empty list here signifies the fact that whatever concept is required to differentiate the failure state from the executable one is not present in the initial concept list* ($\mathbb{C}$). This can be taken as evidence to query the user for more task-related concepts. All locality considerations for sampled states, say focusing on states close to the plan/foil, can be baked into the sampler. The exact algorithm is provided in the appendix.

**Identifying cost function:** We will employ a similar sampling based method to identify the cost function abstraction. Unlike the precondition failure case, there is no single action we can choose but rather we need to choose a level of abstraction for each action in the foil (though it may be possible in many cases to explain the suboptimality of foil by only referring to a subset of actions in the foil). Our approach here would be to find the most abstract representation of the cost function at each step such that of the total cost of the foil becomes greater than that of the specified plan. Thus for a foil $\pi_f = \langle a_1, ..., a_k \rangle$ our objective become

$$\min_{\hat{\mathbb{C}}_1,...,\hat{\mathbb{C}}_k} \Sigma_{i=1..k} \|\hat{\mathbb{C}}_i\| \text{ subject to } \mathcal{C}_s^{abs}(\mathbb{C}_{\pi_f}, \pi_f) > \mathbb{C}(I, \pi)$$

For any given $\hat{\mathbb{C}}_i$, $\mathcal{C}_s^{abs}(\hat{\mathbb{C}}_i, a_i)$ can be approximated by sampling states randomly and finding the minimum cost of executing the action $a_i$ in states containing the concepts $\hat{\mathbb{C}}_i$. We can again rely on a sampling budget to decide how many samples to check and enforce required locality within sampler. Similar to the previous case, *we can identify the insufficiency of the concept set by the fact that we aren't able to identify a valid explanation*. The algorithm can be found in the appendix.

**Confidence over explanations:** Though the methods discussed above are guaranteed to identify the exact model in the limit, the accuracy of the methods is still limited by practical sampling budgets we can employ. For example in the case of precondition failure, at the end of sampling we are not guaranteed that the concept we end up choosing is in fact the true precondition or if generating more samples would have eliminated it. So this means it is important that we are able to establish some level of confidence in the solutions identified. To assess confidence, we will follow the probabilistic relationship between the random variables as captured by Figure 2 (A) for precondition identification and Figure 2 (B) for cost calculation. Where the various random variables captures the following facts: $O_a^s$ - indicates that action $a$ can be executed in state $s$, $c_i \in p_a$ - concept $c_i$ is a precondition of $a$, $O_{c_i}^s$ - the concept $c_i$ is present in state $s$, $\mathcal{C}_s^{abs}(\{c_i\}, a) \geq k$ - the abstract cost function is guaranteed to be higher than or equal to k and finally $O_{\mathcal{C}(s,a) \geq k}$ - stands for the fact that the action execution in the state resulted in cost higher than or equal to $k$. We will allow for inference over these models, by relying on the following simplifying assumptions - (1) the distribution of concepts over the state space is independent of each other, (2) the distribution of all non-precondition concepts in states where the action is executable is the same as their overall distribution across the problem states (which can be empirically estimated), (3) cost distribution of an action over states corre-

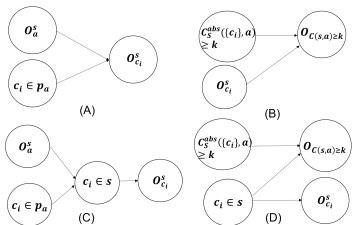

Figure 2: *A simplified probabilistic graphical models for explanation inference, Subfigure (A) and (B) assumes classifiers to be completely correct, while (C) and (D) presents cases with noisy classifier.*

sponding to a concept that does not affect the cost function is identical to the overall distribution of cost for the action (which can again be empirically estimated). The second assumption implies that the likelihood of seeing a non-precondition concept in a sampled state is equal to the likelihood of it appearing in any sampled state (this distribution is denoted as $p_{c_i}$). While the third one implies that for a concept that has no bearing on the cost function for an action, the likelihood that executing the action in a state where the concept is present will result in a cost greater than $k$ will be the same as that of the action execution resulting in a cost greater than $k$ for a randomly sampled state ($p_{\mathcal{C}(.,a) \geq k}$). For a single sample, the posterior probability of explanations for each case can be expressed as follows: For precondition estimation, updated posterior probability for a positive observation can be computed as $P(c_i \in p_a | O_{c_i}^s \wedge O_a^s) = (1 - P(c_i \notin p_a | O_{c_i}^s \wedge O_a^s))$, where

$$P(c_i \notin p_a | O_{c_i}^s \wedge O_a^s) = \frac{p_{c_i} * P(c_i \notin p_a)}{P(O_{c_i}^s | O_a^s)}$$

and for the case of cost function approximation

$$P(\mathcal{C}_s^{abs}(\{c_i\}, a) \geq k | O_{c_i}^s \wedge O_{\mathcal{C}(s,a) \geq k}) = \frac{P(\mathcal{C}_s^{abs}(\{c_i\}, a) \geq k)}{P(\mathcal{C}_s^{abs}(\{c_i\}, a) \geq k)) + p_{\mathcal{C}(.,a) \geq k} * P(\neg \mathcal{C}_s^{abs}(\{c_i\}, a) \geq k))}$$

Full derivation of above formulas can be found in the appendix. The distribution used in the cost explanation, can either be limited to distribution over states where action $a_i$ is executable or allow

for the cost of executing an action in a state where it is not executable to be infinite.

**Using noisy concept classifiers:** Given how unlikely it is to have access to a perfect classifier for any concept, a more practical assumption to adopt could be that we have access to a noisy classifier. However, we assume that we also have access to a probabilistic model for its prediction. That is, we have access to a function $P_\mathbb{C} : \mathbb{C} \to [0, 1]$ that gives the probability that the concept predicted by the classifier is present in the state. Such distributions could be learned from the test set used for learning the classifier. Allowing for noisy observation generally has a more significant impact on the precondition calculation since we can no longer use a single failure (execution of an action in a state where the concept is absent) as evidence for discarding the concept. Though we can still use it as an evidence to update the probability of the given concept being a precondition. We can remove a particular possible precondition from consideration once the probability of it not being a precondition crosses a specified threshold. To see how we can incorporate these probabilistic observations into our confidence calculation, consider the updated relationships presented in Figure 2 (C) and (D) for precondition and cost function approximation respectively. Note that in previous sections, we made no distinction between the concept being part of the state and actually observing the concept. Now we will differentiate between the classifier saying that a concept is present ($O^s_{c_i}$) from the fact that the concept is part of the state ($c_i \in \mathbb{C}(S)$). Now we can use this updated model for calculating the confidence. We can update the posterior of a concept not being a precondition given a negative observation ($O^s_{\neg c_i}$) using the formula

$$P(c_i \notin p_a | O^s_{\neg c_i} \wedge O^s_a) = \frac{P(O^s_{\neg c_i} | c_i \notin p_a \wedge O^s_a) * P(c_i \notin p_a | O^s_a)}{P(O_{\neg c_i} | O^s_a)}$$

Similarly we can modify the update for a positive observation to include the observation model and also do the same for the cost explanation. This can either be empirically calculated from samples with true label or we can assume that this value is going to be approximately equal to the overall distribution of the cost for the action. The derivations for all of the above expressions and formulas for the other cases can be found in appendix. *Note all the experiments reported in this paper were performed using this noisy classifier formulation*. In fact, for Montezuma's revenge we saw that the original formulation couldn't in fact identify the correct concepts.

## 5 EVALUATION

We tested the approach on the open-AI gym's deterministic implementation of Montezuma's Revenge (Brockman et al., 2016) for precondition identification and two modified versions of the gym implementation of Sokoban (Schrader, 2018) for both precondition and cost function identification. Appendix contains the images for all the foils, plans used for each domain.

**Montezuma's Revenge:** We used RAM-based state representation for Montezuma. To add richer preconditions to the settings, we added a wrapper over the original simulators for all the games to render any non-noop action that fails to change the current agent state as an action failure. We selected four invalid foils (generated by the authors by playing the game), three from screen 1 and one from screen 4 of the game. We specified ten concepts for each screen and collected positive and negative examples for each concept by sampling through the state space. We used AdaBoost Classifier (Freund et al., 1999) for concept and had an average accuracy of 99.72%.

**Sokoban Variants:** We used an image-based representation for these domains and had two variations on the standard Sokoban game. One that requires a switch the player could turn on before pushing the box (we will refer to this version as Sokoban-switch) and the second version (Sokoban-cells) included particular cells from which it is costlier to push the box. For Sokoban-switch, we had two variations one in which turning on the switch was a precondition for the action and another one in which it merely reduced the cost of pushing the box. The plan and foil were generated by the authors by playing the game. Additionally, we made sure the plan was part of the optimal policy. *We used a survey to collect the set of concepts for these variants*. The survey allowed participants to interact with the game through a web interface (the cost based version for Sokoban-switch and Sokoban-cell), and at the end, they were asked to specify game concepts that they thought were relevant for particular actions. We received 25 unique concepts from six participants for Sokoban-switch and 38 unique concepts from seven participants for Sokoban-cell. We converted the user descriptions of concepts to scripts for sampling positive and negative instances. Consequently, we focused on 18 concepts and 32 concepts for Sokoban-switch and Sokoban-cell respectively based on the frequency with which they appear in game states and used Convolutional Neural Networks (CNNs) for the classifier. The classifiers had an average accuracy of 99.46% for Sokoban-switch and 99.34% for Sokoban-cell.The exact filtering conditions

provided used along with the network details and hyperparameters are provided in the appendix.

**Explanation identification:** We ran the search for identifying preconditions for Montezuma's foils and Sokoban-switch and cost function identification on both Sokoban. From the collected list of concepts, we doubled the final concept list used by including negations of each concept (so 20 each for Montezuma and 36 and 64 for Sokoban variants). The probabilistic models for each classifier were calculated from the corresponding test sets. For precondition identification, the search was run with a sampling budget of 500 and a cutoff probability of 0.01 for each concept. The search was able to identify the expected explanation for each foil. For Montezuma, the explanations had mean confidence of 0.5044 for foils in screen 1 and a confidence value of 0.8604 for the foil in screen 4. The ones in screen 1 had lower probabilities since they were based on more common concepts and thus their presence in the executable states was not strong evidence for them being a precondition. For Sokoban-switch the confidence was 0.7279. For cost function identification, the search was run with a sampling budget of 750, and all the calculations, including both computing the concept distribution and updating the probability of explanation, were limited to states where the action was executable. Again the search was able to find the expected explanation. We had average confidence of 0.9996 for the Sokoban-switch and 0.998 for the Sokoban-cell.

**User study:** With the basic explanation generation methods in place, we were interested in evaluating if users would find such an explanation helpful. All studies were performed while following local IRB protocols. Screenshots of interface used for the study along with additional details can be found in the appendix. Specifically, the hypotheses we tested were

*Hypothesis 1*: Missing precondition information is a useful explanation for action failures.
*Hypothesis 2*: Abstract cost functions are a useful explanation for foil suboptimality.
*Hypothesis 3*: Concept based precondition explanations help users understand the task better than saliency map based ones.

For H1 and H2, we went with a within-subject study design where the participants were shown an explanation generated by our method along with a simple baseline. For precondition case (H1), the baseline involved pointing out just the failing action and the state it was executed. For the cost case (H2), it involved pointing out the exact cost of executing each action in the foil. The users were asked to choose the one they believed was more useful (the choice ordering was randomized to ensure the results were counterbalanced) and were also asked to report on a Likert scale the completeness of the chosen explanation. For each foil, we took the explanation generated by the search and converted it into text by hand. For H1, we collected 20 replies in total (five per foil) and 19 out of the 20 participants selected precondition based explanation as the choice. On the question of whether the explanation was complete, we had an average score of 3.47 out of 5 on the Likert scale (1 being not at all complete and 5 being complete). For H2, we again collected 20 replies in total (ten per foil) and found 14 out of 20 participants selected the concept-based explanation over the simple one. The concept explanations had on average a completeness score of 3.21 out of 5. The results seems to suggest that in both cases people did prefer the concept-based explanation over the simple alternative. The completeness results suggest that people may like, at least in some cases, to receive more information about the model.

H3 is particularly interesting in the case of precondition failures because under specific settings a Saliency map based explanation could highlight areas corresponding to failed precondition concepts (especially when concepts corresponds to local regions within the image). Though even in such cases the user has to still figure out what concepts these highlighted regions may correspond to and from the highlighted concepts figuring out the one that might be the actual precondition. We measured the user's understanding of the task by their ability to solve the task themselves. Here we went with a between-subject study design. Each participant was allowed to play the precondition variant of the sokoban-switch game. They were asked to finish the game within 3 minutes and were told there would be bonuses for people who finish the game in the shortest time. During the game, if the participant performs an action whose preconditions are not met, then the current episode ends and they have to restart the game. Whenever such an invalid action is executed, then the users are shown an explanation for why the action might fail. One group of users were provided the precondition explanations generated through our method, while the rest were presented with a saliency map generated by using a variant of state of the art saliency-map based explanation method (Greydanus et al. (2018)) and were told that these would be the parts of the state an AI agent would focus on if it was acting in that state. In all the saliency map explanations one of the parts of the state that was highlighted was the switch. The details of the Saliency generation method along with the explanation images shown are provided in the appendix. In total we collected 30 responses, but had to discard six responses because five participants had reported in their submission of not

seeing an explanation and one response was discarded as they refreshed the game in the middle. Out of the remaining 24 responses, 11 corresponded to concept explanations and 13 to saliency based explanations. Participants who got concept based explanation were able to solve the game on average 34.56 steps and 53.56 secs, as opposed to the group who received saliency explanations who took on average 52.92 steps and 71.85 secs. This is in accordance with H3, though we can't establish a statistically significant difference based on confidence intervals (reported in A.6).

## 6 RELATED WORK

There is an increasing number of works investigating the use of high-level concepts to provide meaningful post-hoc explanations to the end-users. The representative works in this direction include works like Bau et al. (2017), TCAV (Kim et al., 2018) and its various offshoots like (Luss et al., 2019) that have focused on one-shot decisions. Most works in explaining sequential decision-making problems have used a model specified in a shared vocabulary as a starting point for explanation or focus on saliency-based explanations (c.f. (Chakraborti et al., 2020)), with very few exceptions. Authors of (Hayes & Shah, 2017) have looked at the use of high-level concepts for policy summaries. They use logical formulas to concisely characterize various policy choices, including states where a specific action may be selected (or not). Unlike our work, they are not trying to answer why the policy ends up choosing a specific action (or not). (Waa et al., 2018) looks at addressing the suboptimality of foils while supporting interpretable features, but it requires the domain developer to specifically encode positive and negative outcomes to each action. In addition to not addressing possible vocabulary differences between a system developer and the end-user, it is also unclear when it is possible to attach negative and positive outcomes to individual actions. Another related work is the approach studied in (Madumal et al., 2020). Here, they are also trying to characterize dynamics in terms of high-level concepts, but assume that the full structural relationship between the various variables is provided upfront. The explanations discussed in this paper can also be seen as a special case of Model Reconciliation explanation (c.f (Chakraborti et al., 2017)), where the human model is considered to be empty. The usefulness of preconditions as explanations has also been studied by works like (Winikoff, 2017; Broekens et al., 2010). Our effort to associate action cost to concepts could also be contrasted to efforts in (Juozapaitis et al., 2019) and (Anderson et al., 2019) which leverage interpretable reward components. Their methods rely on having access to reward functions represented using interpretable components.

## 7 CONCLUSION

We view the approaches introduced in the paper as the first step towards designing more general symbolic explanatory methods for sequential decision-making problems that operate on inscrutable representations. The current methods facilitate generation of explanations in user-specified terms for sequential decisions by allowing users to query the system about alternative plans. We implemented these method in multiple domains and evaluated the effectiveness of the explanation using user studies. While contrastive explanations are answers to questions of the form "Why P and not Q?", we have mostly focused on refuting the foil (the "not Q?" part). This is because, in the presence of a simulator, it is easier to show why the plan is valid by simulating the plan. We can further augment such traces with the various concepts that are valid at each step of the trace. Also, note that the methods discussed in this paper can still be used if the user's questions are specified in term of temporal abstraction over the agent's action space. As long as the system can simulate the foils raised by the user, we can directly use the methods discussed in the paper. While most of the discussions in the paper has been focused on deterministic domains, these ideas also carry over to stochastic domains. The way we are identifying preconditions and estimating cost functions remain the same in stochastic domains. The only difference would be the estimation of the value or the failure point of the foil. One way to easily adapt them to our method would be to compare against the worst case execution cost of the foil or the failure point under one of the execution traces corresponding to the foil. Also while we mostly focused on propositional concepts, our framework is also completely compatible with scenarios where the user may be more comfortable providing relational concepts, since those could be compiled down into propositonal ones. Another avenue of future work could be to see how one could automatically acquire the required concepts, similar for example to the work done by (Ghorbani et al., 2019) in the context of single-shot decisions.

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

## A  APPENDIX

### A.1  APPENDIX OVERVIEW

This appendix contains the following information; (1) the representational assumptions used by the method, (2) pseudo-code for the algorithms used, (3) the derivation of the formulas for confidence calculation (4) the derivation of the formulas for using a noisy classifier and finally (5) the details on the experiment that are not included in the main paper (6) screenshot of the various interfaces.

### A.2  REPRESENTATIONAL ASSUMPTIONS

The central representational assumption we are making is that it is possible to approximate the applicability of actions and cost functions in terms of high-level concepts. Apart from the intuitive appeal of such models (many of these models have their origin in models from folk psychology), these representation schemes have been widely used to model real-world sequential decision-making problems from a variety of domains and have a clear real-world utility (Benton et al., 2019). We agree that there may be problems where it may not be directly applicable, but we believe that this is a sound initial step and is applicable to many domains where currently Reinforcement Learning (RL) based decision-making systems are being successfully used, including robotics and games.

Apart from this basic assumption, we make one additional representational assumption, namely, that the precondition can be expressed as a conjunction of positive concepts. Note that the assumption doesn't restricts the applicability of the methods discussed here. Our framework can still cover cases where the action may require non-conjunctive preconditions. To see why, consider a case where the precondition of action $a$ is expressed as an arbitrary propositional formula, $\phi(\mathbb{C})$. In this case, we can express it in its conjunctive normal form $\phi'(\mathbb{C})$. Now each clause in $\phi'(\mathbb{C})$ can be treated as a new compound positive concept. Thus we can cover such arbitrary propositional formulas by expanding our concept list with compound concepts (including negations and disjuncts) whose value is determined from the classifiers for the corresponding atomic concepts.

### A.3  ALGORITHM

This algorithm for identifying the precondition is specified as Algorithm 1. The algorithm takes

---

**Algorithm 1** Algorithm for Finding Missing Precondition

---

1:  **procedure** PRECONDITION-SEARCH
2:  *Input*:    $s_{\text{fail}}, a_{\text{fail}}, \text{Sampler}, \mathcal{M}_{\text{sim}}, \mathbb{C}, \ell$
3:  *Output*: Missing precondition $C_{\text{prec}}$
4:  *Procedure*:
5:      poss_precondition_set $= \mathbb{C} \setminus \mathbb{C}(s_{\text{fail}})$
6:      sample_count $= 0$
7:      **while** sample_count $< \ell$ **do**
8:          $s \sim \text{Sampler}$
9:          **if** $T(s, a_{\text{fail}}) \neq \bot$ **then**
10:             poss_prec_set $=$ poss_prec_set $\cap \mathbb{C}(s)$
11:         **if** end condition is met **then return** $C_i \in$ poss_prec_set
12:         sample_count += 1
        **return** Any concept $C_i \in$ poss_prec_set

---

as input the failed action ($a_{\text{fail}}$), the simulator, the state at which it was supposed to be executed as per the foil ($s_{\text{fail}}$), a random sampler that returns valid states of the problem (Sampler), the set of all concepts ($\mathbb{C}$) and an upper bound on the number of samples to be explored ($\ell$). Any locality assumptions we want to enforce (like sampling around the plan/foil) can be baked into the sampler, and a simple way to generate such samplers could be by leveraging random walk either from the initial state or states from the foil or the proposed plan. The algorithm starts by initializing the set of possible missing preconditions to all concepts missing from the failure state ($s_{fail}$). Then we start sampling other possible problem states, and we use each new state where the action is executable to reduce our possible precondition list further (since any concept not part of this state can't be a

---

**Algorithm 2** Algorithm for Finding Cost Function

---

1: **procedure** COST-FUNCTION-SEARCH
2: *Input*: $\pi_f, \mathcal{C}_\pi$, Sampler, $\mathcal{M}_{\text{sim}}, \mathbb{C}, \ell$
3: *Output*: $\mathbb{C}_{\pi_f}$
4: *Procedure*:
5:  **for** conc_limit in 1 to $|\mathbb{C}|$ **do**
6:    current_foil_cost = 0
7:    conc_list = []
8:    **for** i in 1 to k (the length of the foil) **do**
9:      $\hat{\mathbb{C}}_i$, min_cost = find_min_conc_set($\mathbb{C}(T(s, \langle a_1, ...a_{i-1}\rangle)), a_i$, conc_limit, $\ell$)
10:      current_foil_cost += min_cost
11:      conc_list.push($\hat{\mathbb{C}}_i$, min_cost)
12:    **if** current_foil_cost $> \mathcal{C}_\pi$ **then return** conc_list
   **return** Signal that the concept list is incomplete

---

possible precondition). If we know that all concepts required to characterize the preconditions are given upfront, then whenever the set of remaining concepts drops to one, we can exit the loop and return the remaining concept. This is because, as per our setting, there must be at least one missing precondition in the $s_{fail}$. Unfortunately, if there are multiple preconditions that are missing in the original failure state or we are not sure whether the given concept set is complete, then we will have to wait until we exhaust the sampling budget.

An algorithm for cost function search is provided in Algorithm 2. The algorithm takes as its input, the foil, the cost of the original plan $\mathcal{C}_\pi$, the problem simulator, the sampler, the concept set $\mathbb{C}$ and the sampling budget $\ell$. The procedure find_min_conc_set, takes the current concept representation of state $i$ in the foil and searches for the subset $\hat{\mathbb{C}}_i$ of the state with the maximum value for $\mathcal{C}_{\mathcal{S}}^{abs}(\hat{\mathbb{C}}_i, a_i)$, where the value is approximated through sampling (with budget $\ell$), and the subset size is upperbounded by conc_limit. Note that this is only a satisficing algorithm and not an optimal one; but we found it to be effective enough for the scenarios we tested.

## A.4 CONFIDENCE CALCULATION

For confidence calculation, we will be relying on the relationship between the random variables as captured by Figure 2 (A) for precondition identification and Figure 2 (B) for cost calculation. Where the various random variables captures the following facts: $O_a^s$ - indicates that action $a$ can be executed in state $s$, $c_i \in p_a$ - concept $c_i$ is a precondition of $a$, $O_{c_i}^s$ - the concept $c_i$ is present in state $s$, $\mathcal{C}_s^{abs}(\{c_i\}, a) \geq k$ - the abstract cost function is guaranteed to be higher than or equal to k and finally $O_{\mathcal{C}(s,a)>k}$ - stands for the fact that the action execution in the state resulted in cost higher than or equal to $k$.

We will allow for inference over these models by relying on the following simplifying assumptions - (1) the distribution of concepts over the state space is independent of each other, (2) the distribution of all non-precondition concepts in states where the action is executable is the same as their overall distribution across the problem states (which can be empirically estimated) and (3) cost distribution for an action over states corresponding to a concept that does not affect the cost function is identical to the overall distribution of cost for that action (which can again be empirically estimated). The second assumption implies that you are as likely to see a non-precondition concept in a sampled state where the action is executable as the concept was likely to appear at any sampled state (denoted as $p_{c_i}$). While the third one implies that for a concept that has no bearing on the cost function for an action, the likelihood that executing the action in a state where the concept is present will result in a cost greater than $k$ will be the same as that of the action execution resulting in a cost greater than or equal to $k$ for any randomly sampled state ($p_{\mathcal{C}(.,a) \geq k}$).

For a single sample, the posterior probability of explanations for each case can be expressed as follows: For precondition estimation, updated posterior probability for a positive observation can be computed as $P(c_i \in p_a | O_{c_i}^s \wedge O_a^s) = (1 - P(c_i \notin p_a | O_{c_i}^s \wedge O_a^s))$, where

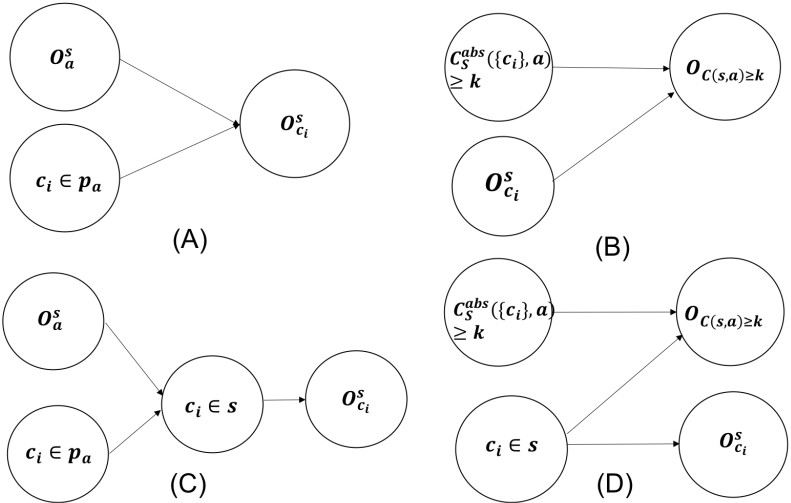

Figure 3: A simplified probabilistic graphical models for explanation inference, Subfigure (A) and (B) assumes classifiers to be completely correct, while (C) and (D) presents cases where the classifier may be noisy.

$$P(c_i \notin p_a | O_{c_i}^s \wedge O_a^s)$$
$$= \frac{P(O_{c_i}^s | c_i \notin p_a \wedge O_a^s) * P(c_i \notin p_a | O_a^s)}{P(O_{c_i}^s | O_a^s)}$$

Given $c_i \notin p_a$ is independent of $O_a^s$ and expanding the denominator we get

$$= \frac{P(O_{c_i}^s | c_i \notin p_a \wedge O_a^s) * P(c_i \notin p_a)}{\begin{array}{c} P(O_{c_i}^s | c_i \notin p_a \wedge O_a^s) * P(c_i \notin p_a) + \\ P(O_{c_i}^s | c_i \in p_a \wedge O_a^s) * P(c_i \in p_a) \end{array}}$$

From our assumption, we know $P(O_{c_i}^s | c_i \notin p_a \wedge O_a^s)$ is same as the distribution $c_i$ over the problem states ($p_{c_i}$) and $P(O_{c_i}^s | c_i \in p_a \wedge O_a^s)$ must be one.

$$= \frac{p_{c_i} * P(c_i \notin p_a)}{p_{c_i} * P(c_i \notin p_a) + P(c_i \in p_a)}$$

For cost calculation, we have

$$P(\mathcal{C}_s^{abs}(\{c_i\}, a) \geq k | O_{c_i}^s \wedge O_{\mathcal{C}(s,a) \geq k}) = \frac{P(O_{\mathcal{C}(s,a) \geq k} | O_{c_i}^s \wedge \mathcal{C}_s^{abs}(\{c_i\}, a) \geq k) * P(\mathcal{C}_s^{abs}(\{c_i\}, a) \geq k | O_{c_i}^s)}{P(O_{\mathcal{C}(s,a) \geq k} | O_{c_i}^s)}$$

Where $P(O_{\mathcal{C}(s,a) \geq k} | O_{c_i}^s, \mathcal{C}_s^{abs}(\{c_i\}, a) \geq k)$ should be 1 and $\mathcal{C}_s^{abs}(\{c_i\}, a) \geq k$ independent of $O_{c_i}^s$. Which gives

$$= \frac{P(\mathcal{C}_s^{abs}(\{c_i\}, a) \geq k)}{P(O_{\mathcal{C}(s,a) \geq k} | O_{c_i}^s)}$$

$$= \frac{P(\mathcal{C}_s^{abs}(\{c_i\}, a) \geq k)}{\begin{array}{c} P(O_{\mathcal{C}(s,a) \geq k} | O_{c_i}^s, \mathcal{C}_s^{abs}(\{c_i\}, a) \geq k)) * P(\mathcal{C}_s^{abs}(\{c_i\}, a) \geq k)) + \\ P(O_{\mathcal{C}(s,a) \geq k} | O_{c_i}^s \wedge \neg \mathcal{C}_s^{abs}(\{c_i\}, a) \geq k)) * P(\neg \mathcal{C}_s^{abs}(\{c_i\}, a) \geq k)) \end{array}}$$

From our assumptions, we have $P(O_{\mathcal{C}(s,a) \geq k} | O_{c_i}^s \wedge \neg \mathcal{C}_s^{abs}(\{c_i\}, a) \geq k)) = p_{\mathcal{C}(.,a) \geq k}$

$$= \frac{P(\mathcal{C}_s^{abs}(\{c_i\}, a) \geq k)}{P(\mathcal{C}_s^{abs}(\{c_i\}, a) \geq k)) + p_{\mathcal{C}(.,a) \geq k} * P(\neg \mathcal{C}_s^{abs}(\{c_i\}, a) \geq k))}$$

## A.5 Using Noisy Concept Classifiers

Note that in previous sections, we made no distinction between the concept being part of the state and actually observing the concept. Now we will differentiate between the classifier saying that a concept is present ($O_{c_i}^s$) is a state from the fact that the concept is part of the state ($c_i \in \mathbb{C}(S)$). The relationship between the random variables can be found in Figure 3 (C) and (D). We will assume that the probability of the classifier returning the concept being present is given by the probabilistic confidence provided by the classifier. Of course, this still assumes the classifier's model of its prediction is accurate. However, since it is the only measure we have access to, we will treat it as being correct. Now we can use this updated model for calculating the confidence. For the precondition estimation, we can update the posterior of a concept being a precondition given a negative observation ($O_{\neg c_i}^s$) using the formula

$$P(c_i \notin p_a | O_{\neg c_i}^s \wedge O_a^s) = \frac{P(O_{\neg c_i}^s | c_i \notin p_a \wedge O_a^s) * P(c_i \notin p_a | O_a^s)}{P(O_{\neg c_i}^s | O_a^s)}$$

Where $P(c_i \notin p_a | O_a^s) = P(c_i \notin p_a)$ and we can expand $P(O_{\neg C_i}^s | c_i \notin p_a \wedge O_a^s)$ as follows

$$P(O_{\neg c_i} | c_i \notin p_a \wedge O_a^s) =$$
$$P(O_{\neg c_i} | c_i \in \mathbb{C}(s)) * P(C_i \in \mathbb{C}(s) | c_i \notin p_a \wedge O_a^s) +$$
$$P(O_{\neg c_i} | c_i \notin \mathbb{C}(s)) * P(c_i \notin \mathbb{C}(s) | c_i \notin p_a \wedge O_a^s)$$

Where as defined earlier $P(c_i \notin \mathbb{C}(s) | c_i \notin p_a \wedge O_a^s)$ and $P(c_i \in \mathbb{C}(s) | C_i \notin p_a \wedge O_a^s)$ would correspond to $(1 - p_{c_i})$ and $p_{c_i}$. The denominator also needs to be marginalized over $c_i \notin \mathbb{C}(s)$.

Similarly for posterior calculation for positive observations, we have

$$P(O_{c_i}^s | c_i \notin p_a \wedge O_a^s) =$$
$$P(O_{c_i} | c_i \in \mathbb{C}(s)) * P(c_i \in \mathbb{C}(s) | c_i \notin p_a \wedge O_a^s) +$$
$$P(O_{c_i} | c_i \notin \mathbb{C}(s)) * P(c_i \notin \mathbb{C}(s) | c_i \notin p_a \wedge O_a^s)$$

Now for the cost, we can similarly incorporate the observation model as follows.

$$P(\mathcal{C}_s^{abs}(\{c_i\}, a) \geq k | O_{c_i}^s \wedge O_{\mathcal{C}(s,a)>=k}) = \frac{\begin{array}{c} P(O_{\mathcal{C}(s,a) \geq k}, O_{c_i}^s | \mathcal{C}_s^{abs}(\{c_i\}, a) \geq k) \\ * P(\mathcal{C}_s^{abs}(\{c_i\}, a) \geq k) \end{array}}{P(O_{\mathcal{C}(s,a) \geq k}, O_{c_i}^s)}$$

$$= \frac{\begin{array}{c} (P(O_{\mathcal{C}(s,a) \geq k}, O_{c_i}^s | c_i \in \mathbb{C}(s), \mathcal{C}_s^{abs}(\{c_i\}, a) \geq k) * P(c_i \in \mathbb{C}(s)) + \\ P(O_{\mathcal{C}(s,a) > k}, O_{c_i}^s | c_i \notin \mathbb{C}(s), \mathcal{C}_s^{abs}(\{c_i\}, a) \geq k) * P(c_i \notin \mathbb{C}(s))) * P(\mathcal{C}_s^{abs}(\{c_i\}, a) \geq k) \end{array}}{P(O_{\mathcal{C}(s,a) \geq k}, O_{c_i}^s)}$$

Given their parents, $O_{\mathcal{C}(s,a) \geq k}$ and $O_{c_i}^s$ are conditionally independent, and given its parent $O_{c_i}^s$ is independent of $\mathcal{C}_s^{abs}(\{c_i\}, a) \geq k$), there by giving

$$= \frac{\begin{array}{c} (P(O_{\mathcal{C}(s,a) \geq k} | c_i \in \mathbb{C}(s), \mathcal{C}_s^{abs}(\{c_i\}, a) \geq k) * P(O_{c_i}^s | c_i \in \mathbb{C}(s)) * P(c_i \in \mathbb{C}(s)) + \\ P(O_{\mathcal{C}(s,a) \geq k} | c_i \notin \mathbb{C}(s), \mathcal{C}_s^{abs}(\{c_i\}, a) \geq k) * P(O_{c_i}^s | c_i \notin \mathbb{C}(s)) * P(c_i \notin \mathbb{C}(s))) * P(\mathcal{C}_s^{abs}(\{c_i\}, a) \geq k) \end{array}}{P(O_{\mathcal{C}(s,a) \geq k}, O_{c_i}^s)}$$

Now $P(O_{\mathcal{C}(s,a) \geq k} | c_i \in \mathbb{C}(s), \mathcal{C}_s^{abs}(\{c_i\}, a) \geq k) = 1$ and $P(O_{\mathcal{C}(s,a) \geq k} | c_i \notin \mathbb{C}(s), \mathcal{C}_s^{abs}(\{c_i\}, a) \geq k)$ can either be empirically estimated from true labels or we can make the assumption that is equal to $p_{\mathcal{C}(.,a) \geq k}$ (which we made use of in our experiments), which would take us to

$$= \frac{\begin{array}{c} (P(O_{c_i}^s | c_i \in \mathbb{C}(s)) P(c_i \in \mathbb{C}(s)) + \\ p_{\mathcal{C}(.,a) \geq k} * P(O_{c_i}^s | c_i \notin \mathbb{C}(s)) * P(c_i \notin \mathbb{C}(s))) * P(\mathcal{C}_s^{abs}(\{c_i\}, a) \geq k) \end{array}}{P(O_{\mathcal{C}(s,a) \geq k}, O_{c_i}^s)}$$

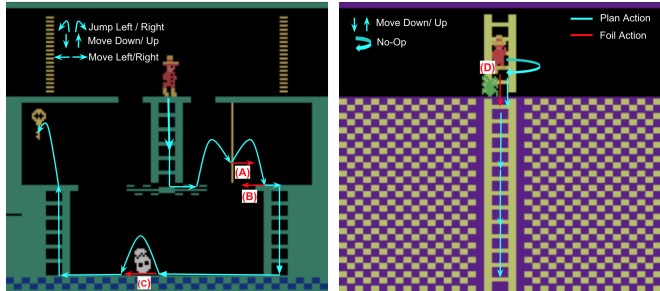

Figure 4: Montezuma Foils: Left Image shows foils for level 1, (A) Move right instead of Jump Right (B) Go left over the edge instead of using ladder (C) Go left instead of jumping over the skull. Right Image shows foil for level 4, (D) Move Down instead of waiting.

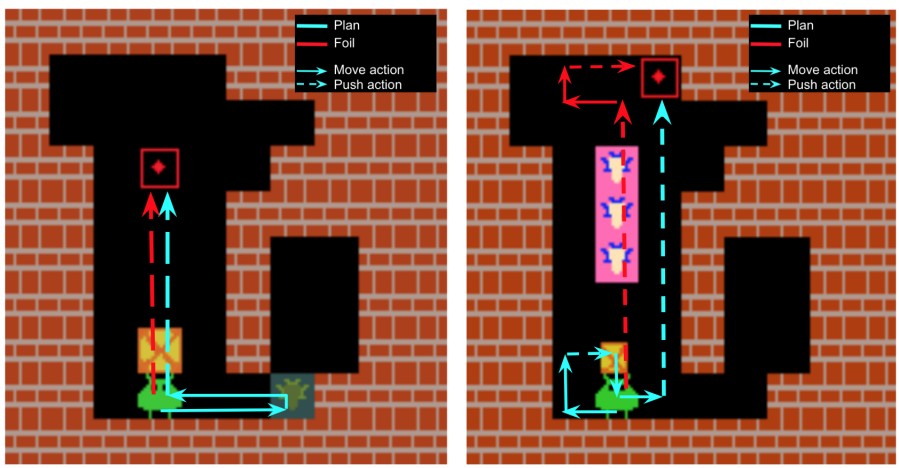

Figure 5: Sokoban Foils: Left Image shows foils for Sokoban-switch, note that the green cell will turn pink once the agent passes it. Right Image shows foil for Sokoban-cell.

A.6    EXPERIMENT DETAILS

For validating the soundness of the methods discussed before, we tested the approach on the open-AI gym implementation of Montezuma's Revenge Brockman et al. (2016) and variants of Sokoban Schrader (2018). Most of the search experiments were run on an Ubuntu 14.0.4 machine with 64 GB RAM.

For Montezuma, we used the deterministic version of the game with the RAM-based state representation (the game state is represented by the RAM value of the game controller, represented by 256-byte array). We considered executing an action in the simulator that leads to the agent's death (falling down a ledge, running into an enemy) or a non-NOOP (NOOP action is a specific agent action that is designed to leave agent's state unchanged) action that doesn't alter the agent position (trying to move left on a ladder) as action failures. We selected four possible foils for the game (illustrated in Figure 4), three from level 1 and one from level 4. The base plan in level 1 involves the agent reaching the key, while level 4 required the agent to cross the level.

For Sokoban, we considered two variants with a single box and a single target. Both variants allow for 8 actions and a NOOP action. Four of those actions are related to the agent's movements in the four directions and four to pushing in specific directions. We restricted the move actions only to be able to move the agent if the cell is empty, i.e., it won't move if there is a box or a wall in that direction. The push action also moves the agent in the direction of push if there is a box in that direction, and the agent will occupy the cell previously occupied by the box (provided there are no walls to prevent the box from moving). Similar to Montezuma, we consider any action that

doesn't change the agent position to be a failure. The first version of Sokoban included a switch the player could turn on to push the box (we will refer to this version as Sokoban-switch), and the second version (Sokoban-cells) included particular cells from which it is costlier to push the box. We considered two versions of Sokoban-switch, one in which turning on the switch only affected the cost of pushing the box and another one in which it was a precondition. For the cost case of Sokoban-switch, while the switch is on (i.e. the cell is pink), all actions have unit cost, while when the switch is off, the cost of pushing actions is 10. The cell can be switched on by visiting it, and any future visit will cause it to turn off. For the precondition-version, pushing of the box while the switch is off causes the episode to end with high cost (100). Since we also trained an RL agent for this version for generating the saliency map. We also added a small penalty for not pushing the switch and a penalty proportional to the distance between the box and the target. In Sokoban-cells, the cost of all actions except pushing boxes in the pink region is one, while that of pushing boxes in the pink region is 10. We selected one foil per variation, and the original plan and the foil are shown in Figure 5.

**Concept Learning** For Montezuma, we came up with ten base concepts for each level that approximates the problem dynamics at the level, including the foil failure. We additionally created ten more concepts by considering the negations of them. All state samples (used to generate the samples for the classifier and the algorithm) were created by randomly selecting one of the states from the original plan and then performing random walks from the selected state. For the classifiers, we used game-specific logic and RAM byte values to identify each positive instance and then randomly selected a set of negative examples. We used around 600 positive examples (except for the concepts $skull\_on\_right$ and $skull\_on\_left$ in level 1, which had 563 and 546 examples, respectively) and twice as many negative examples for each concept. These RAM state examples were fed to a binary AdaBoost Classifier Freund et al. (1999) (Scikit-learn implementation Pedregosa et al. (2011) version 0.22.1, with default parameters), with 70% of samples used as train set and the rest as the test set, for each concept. Finally, we obtained a test accuracy range of 98.57% to 100%, with an average of 99.72% overall concepts of both the levels. All the samples used for the classifier were collected from 5000 sampling episodes for level 1 and 4000 sampling episodes for level 4. During the search, We used a threshold of 0.55 on classifiers for concepts of level 1, such that a given state has a given concept when the classifier probability is greater than 0.55, to reduce false positives. The code for sampling and training the classifiers can be found in the directory PRECOND_BLACKBOX/sampler_and_conceptTrain inside the code directory.

For the Sokoban variants, we wanted to collect at least the list of concepts from people. We used surveys to collect the set of concepts. The survey allowed participants to interact with the game through a web interface, and at the end, they were asked to specify game concepts that they thought were relevant for particular actions. Each user was asked to specify a set of concepts that they thought were relevant for four actions in the game. They were introduced to the idea of concepts and their effect on the action by using PACMAN as an example and presenting three example concepts. The exact instructions and screenshots of the interface used for Sokoban-cell can be found in the file Sokoban_cell_survey.pdf in the directory Study_Website_Pdfs, which is part of the supplementary file zip. For Sokoban-switch, we collected data from six participants, four of whom were asked to specify concepts for push actions and two people for move actions. For Sokoban-cell, we collected data from seven participants, six of whom were asked to specify concepts for push actions, and one was asked to specify concepts for move action. We went through the submitted concepts and clustered them into unique concepts using their description. We skipped ones where they just listed strategies rather than concepts describing the state. We removed two concepts from the Sokoban-cell list and two from Sokoban-switch because we couldn't make sense of the concept being described there. For Sokoban-switch, we received 25 unique concepts, and for Sokoban-cell, we collected 38 unique concepts. We wrote scripts for each of the concepts and used it to sample example states. We ran the sampler for 1000 episodes to collect the examples for the concepts. We trained classifiers for each of the concepts that generated more than 10 positive examples for the concepts. For sokoban-switch, we removed two additional concepts because their training set didn't contain any positive examples. We had, on average, 178.46 positive examples for Sokoban-cell per concept and 215.55 for Sokoban-switch. We used all the other samples as negative examples. We again used 70% of samples for training and the remaining for testing. We used Convolutional Neural Networks (CNNs) based classifiers for the Sokoban variants. The CNN architecture involved four convolutional layers, followed by three fully connected layers that give a binary classification output. The

average accuracy of the Sokoban-switch was 99.46%, and Sokoban-cell was 99.34%. The code used for sampling and training for each domain can be found under the folder COST_TRAINER (inside the directory BLACKBOX_CODE). The classifier network is specified in the file CNNnetwork.py. The details on how to run them are provided in the README file in the root code directory.

**Explanation Identification:** For Montezuma, the concept distribution was generated using 4000 episodes, and the probability distribution of concepts ranged from 0.0005 to 0.206. For some of the less accurate models, we did observe false negatives resulting in the elimination of the accurate preconditions and empty possible precondition set. So we made use of the probabilistic version of the search with observation probabilities calculated from the test set. We applied a concept cutoff probability of 0.01, and in all cases, the precondition set reduced to one element (which was the expected precondition) in under the 500 step sampling budget (with the mean probability of 0.5044 for foils A, B & C. Foil D, in level 4, gave a confidence value of 0.8604). The ones in level 1 had lower probabilities since they were based on more common concepts, and thus, their presence in the executable states was not strong evidence for them being a precondition.

For each Sokoban variant, we ran another 1000 episode sampler, which used random restarts from the foil states to collect samples that we used for the explanation generation. During the generation of the samples, we used the previously learned classifiers to precompute the mapping from concepts to states.

We used a variant of Algorithm 2, where we sped up the search by allowing for memoization. Specifically, when sampling is done for an action and a specific conc_limit for the first time, then we precompute the min-cost for all possible concept subset of that size. Then for every step that uses that action, we look up the min value for the subset that appears in the state. The search was run with a sampling budget of 750. For calculating the confidence, all required distributions were calculated only on the states where the action was executable. Again the search was able to find the expected explanation. We had average confidence of 0.9996 for the Sokoban-switch and 0.998 for the Sokoban-cell. The exact observation models values used can be found in constant.py under the directory COST_BLACKBOX/src, and the file cost_explainer.py in the same directory contains the code for the exact search we used.

**User study:** With the basic explanation generation method in place, we were interested in evaluating if users would find such an explanation helpful. Specifically, the hypotheses we tested were *Hypothesis 1: Missing precondition information is a useful explanation for action failures.*

*Hypothesis 2: Abstract cost functions are a useful explanation for foil suboptimality.*

*Hypothesis 3: Concept based-explanation (at least precondition ones) help users understand the task better than Saliency map based ones.*

To evaluate this, we performed a user study with all the foils used along with the generated explanation and a simple baseline. In the study, each participant was presented with a random subset of the concept we used for the study (around five) and then was shown the plan and a possible foil. Then the participant was shown two possible explanations for the foil (the generated one and the baseline) and asked to choose between them. There were additional questions at the end asking them to specify what they believed was the completeness of the selected explanation, on a Likert scale from 1 to 5 (1 being least complete and 5 being the most). They were also provided a free text field to provide any additional information they felt would be useful.

For H1, the options showed to the user includes one that showed the state at which the foil failed along with the information that the action cannot be executed in that state and the other one reported that the action failed because a specific concept was missing (the order of the options was randomized). In total, we collected data from 20 participants, where 7 were women, the average age was 25, and 10 people had taken an AI class. We found that 19 out of the 20 participants selected precondition based explanation as a choice. On the question of whether the explanation was complete, we had an average score of 3.476 out of 5 on the Likert scale (1 being not at all complete and 5 being fully complete). The results seem to suggest that information about missing precondition are useful explanations though these may not be complete. While not a lot of participants provided information on what other information would have been useful, the few examples we had generally

Figure 6: Saliency map based explanation shown to users as part of H3

pointed to providing more information about the model (for example, information like what actions would have been possible in the failed state).

For H2, the baseline involved pointed out the exact cost of executing each action in the foil, and concept-explanation showed how certain concepts affected the action costs. In the second case, all action costs were expressed using the abstract cost function semantics in that they were expressed as the 'action costing at least X' even though in our case, the cost was the same as the abstract cost. For H2, again, we collected 20 replies in total (ten per foil) and found 14 out of 20 participants selected the concept-based explanation over the simple one. The concept explanations had, on average, a completeness score of 3.214 out of 5. The average age of the participants was 24.15, 10 had AI knowledge out of 20 people, 11 were masters students (rest were undergrad), and we had 14 males, 5 females, and one participant did not specify.

For H3, as mentioned the baseline was a saliency map based explanation. For generating the saliency map, we trained the RL agent using DQN with prioritized experience replay (Schaul et al., 2015)[2]. The agent was trained for 420k epochs. The Saliency map itself was generated for four states, with the agent placed on the four sides of the box. The saliency map itself was generated using Greydanus et al. (2018), where we used only the procedure for generating the map for the critic network[3]. Figure 6 shows the saliency map generated for one of the images. This was shown when the user tried push up action and fails. At the beginning of the study, both groups of the users were made to familiarize themselves with five concepts that were randomly ordered (the concepts themselves remained the same) and had to take a quiz matching new states to those concepts, before moving on to play the game. Out of the 24 responses we considered, six identified as female and 16 identified as men. Nine of the participants reported they had some previous knowledge of AI, but only one participant reported having any planning knowledge. *For H3, the Participants who got concept based explanation were able to solve the game on average 34.56 steps and 53.56 secs (with a 95% confidence interval of $\pm 12.56$ and $\pm 23.21$ respectively), as opposed to the group who received saliency explanations took on average 52.92 ($\pm 16.01$) steps and 71.85 ($\pm 28.65$) secs. Note that given the overlap between the confidence intervals we can't establish a statistically significant difference between the conditions using the intervals.*

PDF files showing the screenshots of the user study website for a scenario for the precondition explanation test and one from cost explanation can be found in User-study-precondition.pdf and User-study-cost.pdf in the directory Study_Website_Pdfs. The actual data collected from the user studies and the concept survey can be found in the directory USER_STUDY_FEEDBACK. But below, we have included some screenshots of the interface

---

[2]For exact agent, we followed the approach described in `https://github.com/higgsfield/RL-Adventure/blob/master/4.prioritized%20dqn.ipynb`

[3]We made use of the code available at `https://github.com/greydanus/visualize_atari`

| Action | Concept with Max difference | Max Absolute Difference in Estimates | Average Difference in Estimates |
|---|---|---|---|
| push up | empty_above | 0.018 | 0.004 |
| push down | empty_below | 0.0154 | 0.0033 |
| push left | empty_below | 0.0163 | 0.0037 |
| push right | empty_below | 0.0172 | 0.0046 |
| move up | empty_left | 0.002 | 0.0009 |
| move down | wall_left | 0.0017 | 0.0007 |
| move left | empty_right | 0.0032 | 0.0009 |
| move right | empty_right | 0.0045 | 0.0008 |

Figure 7: Results from Sokoban-switch on the distribution of non-precondition concepts for each action. For each action the table reports the concept with the maximum difference between the distribution of concept for that specific version, versus the overall distribution and the average difference in estimates across concepts.

## A.7 ANALYSIS OF ASSUMPTIONS MADE FOR CONFIDENCE CALCULATIONS

In this section we present results from additional tests we ran to verify some of the assumptions made in the confidence calculations on the Sokoban variants. We mainly focused on Sokoban variants since the concepts were collected directly from users and we tested the assumptions - (1) the distribution of all non-precondition concepts in states where the action is executable is the same as their overall distribution across the problem states (which can be empirically estimated) and (2) cost distribution of an action over states corresponding to a concept that does not affect the cost function is identical to the overall distribution of cost for the action (which can again be empirically estimated). We didn't run a separate test on the independence of concepts as we saw that many of the concepts listed by the users were in fact correlated and were denoting similar or even the same phenomena. All concepts were assumed to be distributed according to a Bernoulli distribution, whose MLE estimates were calculated by running the sampler for ten thousand episode, where we used states from the original successful/optimal plan as the initial state for the random walk (ensuring the distributions are generated from state space local to the plan of interest). For first assumption, we compared the distribution of concept for states where the action was executed against the distribution of the concept over all the sampled states. For the second assumption, we compared the distribution of the states with the high cost ($>=10$) where the concept is present versus the distribution of high cost for the action.

Table 7, summarizes the results from testing the first assumption for Sokoban-switch. For each action the table reports the non-precondition concept which had the maximum difference in estimates (the reported difference in the table). In this domain, the only precondition concept is switch_on for the push actions. As we can see for the domain, the differences are pretty small and we expect the differences to further reduce once we start accounting the correlation between concepts.

Table 8 presents the results for the second assumption. In the cases of Sokoban-switch, we again skipped the switch_on concept and for Sokoban-Cell we skipped the concepts related to the pink cells since they are all highly correlated to central concept controlling the cost function (on_pink_cell).

| Domain | Action | Concept with Max difference | Max Absolute Difference in Estimates | Average Difference in Estimates |
|---|---|---|---|---|
| Sokoban-Switch | push up | wall_left_below_of_box | 0.1132 | 0.0461 |
| | push down | wall_left_below_of_box | 0.1107 | 0.0473 |
| | push left | above_switch | 0.1135 | 0.0461 |
| | push right | wall_left_below_of_box | 0.111 | 0.0479 |
| Sokoban-Cell | push up | box_on_right | 0.0956 | 0.0411 |
| | push down | box_on_right | 0.1098 | 0.0476 |
| | push left | box_on_right | 0.1012 | 0.0486 |
| | push right | wall_on_left | 0.0889 | 0.0433 |

Figure 8: Results from Sokoban-switch and Sokoban-Cell on the distribution of action cost across different concepts. Here we report only the cost for push actions, since only those actions result in higher cost.

## A.8 STUDY INTERFACE SCREENSHOTS

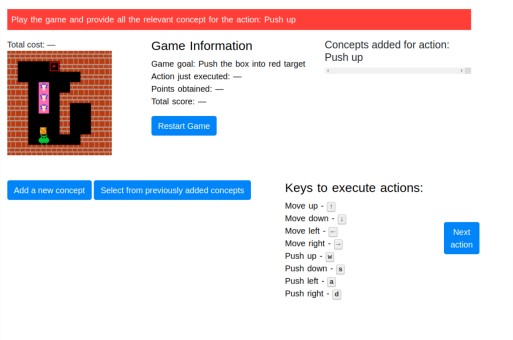

Figure 9: Screenshot from the survey done to collect sokoban concepts.

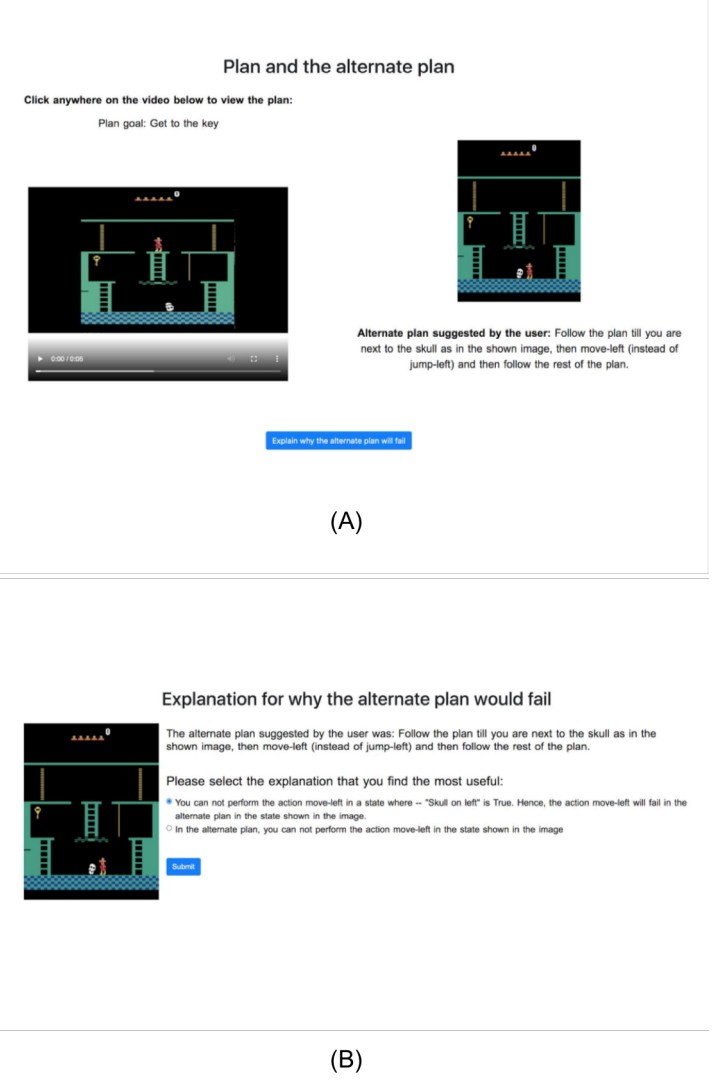

Figure 10: Screenshot from the study interface for H1.

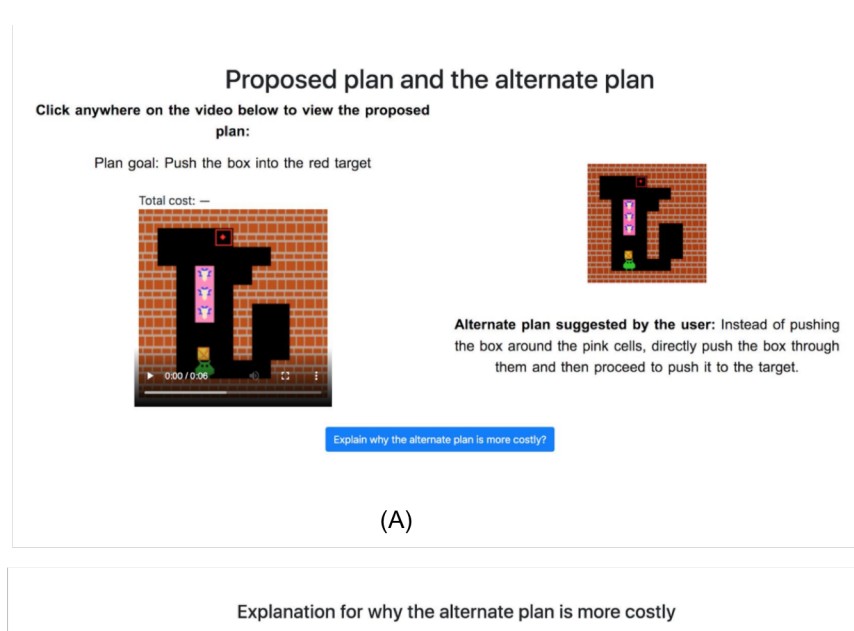

(A)

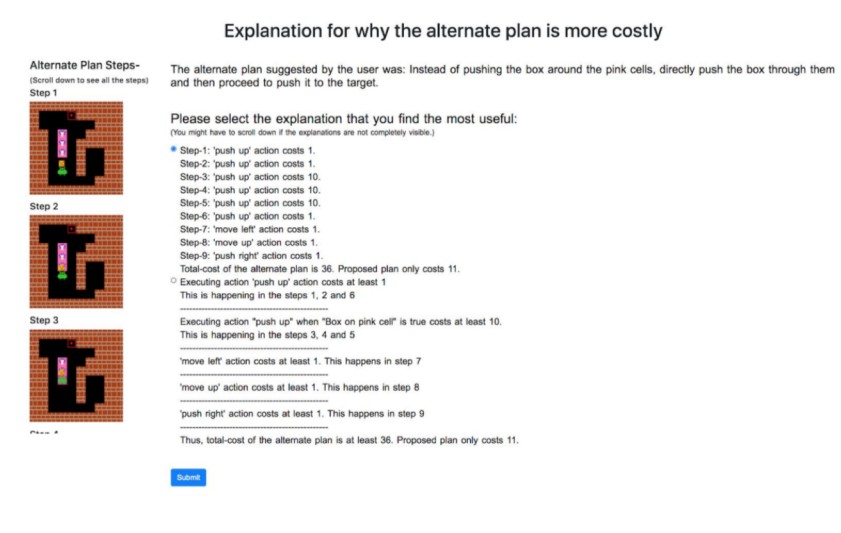

(B)

Figure 11: Screenshot from the study interface for H2.

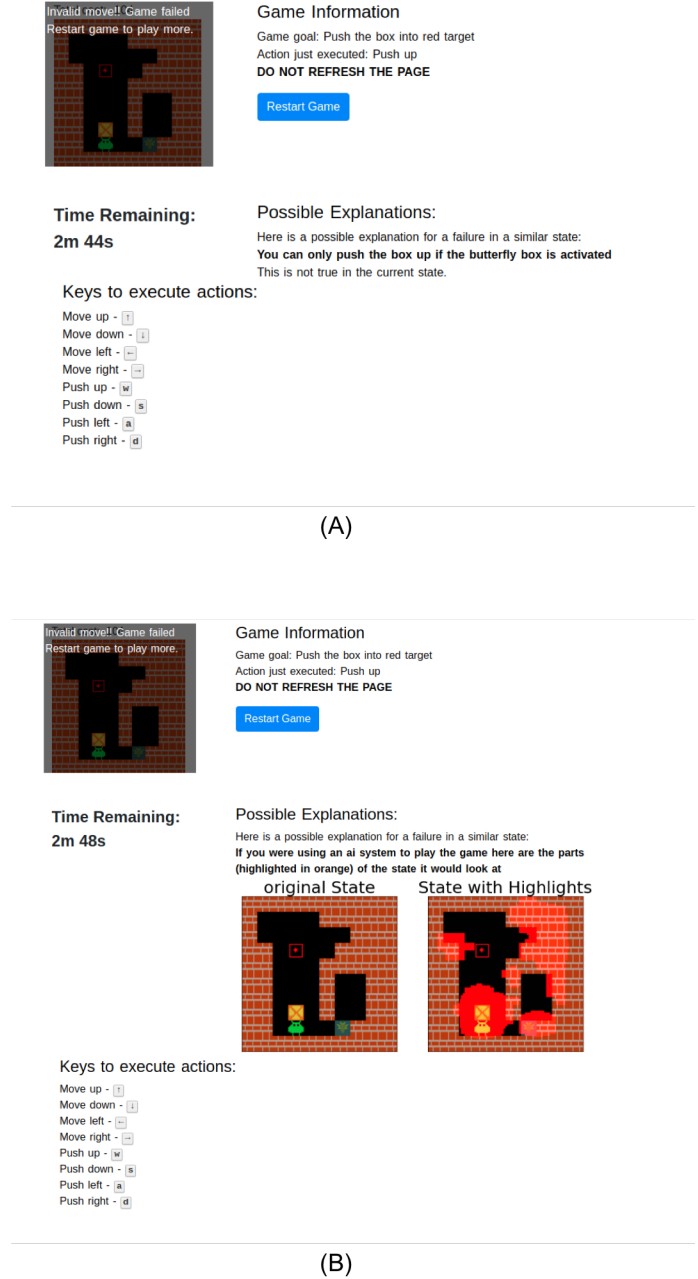

Figure 12: Screenshot from the study interface for H2.

