# OpenReview forum: "Bridging the Gap: Providing Post-Hoc Symbolic Explanations for Sequential Decision-Making Problems with Inscrutable Representations"
_ICLR.cc/2021/Conference — Reject_

### Official Review · AnonReviewer2 · 2020-10-17
**Interesting explanation method; minor concerns in user study**

**Rating:** 7
**Confidence:** 4

**Review:**

#### Summary
This paper introduces a new method for contrastive explanation of symbolic models on sequential decision-making problems, i.e. explaining why a foil plan is not as good as the system proposed plan. The suboptimality of foil plans is categorized into two types: invalid actions and larger cost.

To make the explanation more understandable to human, the author introduces "concepts" to explanation, which are assumed propositional properties of states. Concept classifiers can be trained on samples to predict the presence of each concept in a state. To explain an invalid action, the method reports a missing precondition concept of the failing action; to explain a larger cost, it reports a set of representative concepts such that the foil actions under these concepts are guaranteed to give a larger cost than the proposed plan. The paper also provides detailed algorithms to find preconditions or representative concepts (using "abstract cost functions") for actions. The authors also introduce simple PGMs to evaluate the confidence of each explanation.

The authors conducted user studies on the proposed explanation method and demonstrated its usefulness.

#### Strength
- Model explanation is indeed a very important field today for any ML models. Sequential decision-making models are popular and widely used, but explanation methods for them are still somewhat underexplored. As reported in the paper, previous work either includes "concepts" directly in model learning (not post-hoc) or use saliency maps (not human concepts); this work introduces human concepts to post-hoc explanation.
- The whole explanation method is novel and makes sense to me. The ideas are intuitive, but the algorithms for finding the explanations and evaluating the confidence are non-trivial.

#### Weakness
- In my opinion, the user study still has room for improvement.
	- For H1 and H2, directly asking the user "which one is more useful" does not look like a good way of comparison. At least to me, it's hard to tell which one is more useful if I have already read both and understood. Maybe it's better to only show one explanation to a participant, and ask how well they understood, or let them perform a task as in H3.
	- For H1, as I see in Figure 8B, the explanation with concepts (CE) has exactly more information than the baseline explanation (BE). In other words, CE is concept + image while BE is only image; BE is a strict subset of CE. Such a comparison does not seem very useful. Would it be better if you, for example, compare concept-only with image-only, such as removing "in the state shown in the image" in CE?
	- For H3, the task setting is interesting, but in the shown example, the concept that the user has to learn is merely to use the switch, which looks somewhat too simple and not very interesting. It would be more attractive if the concept is more complex (e.g. you can't fall off a plane or touch an enemy as in Montezuma).
	- I think the participants' background information (age, gender, how recruited, etc.) is important for a user study and I suggest having it in the main paper or at least making it easier to notice.
- It seems that empirical studies are not provided for confidence score calculation, i.e. how well do the scores correspond with actual explanation accuracy.

#### Questions
- In Section 4 line 16, you mentioned "In cases where we are guaranteed that the concept list is exhaustive..." Could you provide some examples of these cases?

- In Section 5 "Explanation identification", you mentioned "The search was able to identify the expected explanation for each foil." Does that mean all the output explanations are accurate? If so, it will be interesting to see how the search performs on more complex tasks.
- For explanations on larger cost, as in Figure 9B, to me the most ideal explanation would be just *Executing action "push up" when "Box on pink cell" costs at least 10*; other lines are not useful. In other words, it might be better to focus on steps where the foil has larger cost than proposed plan. Do you have any thoughts on this line?

#### Typos
- Section 2: symbol $G$ (set of goal states) font inconsistent
- Appendix A.3 Algorithm 2 line 12: symbol broken

---

> ### Author Response · Authors · 2020-11-11
> **Response to Review**
>
> Showing both explanations to participants for H1 and H2:
> Through H1 and H2 we were trying to perform a subjective evaluation of the explanation content and through H3 an objective evaluation. As such we went with a within-subject study design for H1 and H2, i.e. the same user is shown both conditions so that the subjective responses comparing the two methods aren't too affected by individual variations. But we did counterbalance the study, i.e., randomized the order in which the conditions are shown to avoid any biasing that could happen from the order of the explanations.
>
> Concept plus image:
> Given the fact that the same participants are exposed to both types of information, we thought mentioning the image along with concepts would help us get more uniform responses from all the users. Otherwise we might have cases where some users might make use of both image and concept and others who might not. Though to clarify within explanatory literature, additional information is not always considered to be more helpful. In fact one of the core properties of useful explanation is minimality and we believe the users would have penalized the explanation if they thought the concept explanation was providing redundant information.
>
>
> Explanation confidence accuracy:
> We don't have access to the ground truth information on explanation confidence, but we found that in the cases we tested the highest confidence explanation corresponded to precondition concepts.
>
> Questions:
>
> Q1.  "In cases where we are guaranteed that the concept list is exhaustive..." Could you provide some examples of these cases?
>
> A common example where we could have confidence that we have the exhaustive list could be cases where the concepts are compiled from a large number of users who are familiar with the domain. Another case could be when the concepts are generated from textual descriptions of the actual domain.
>
> Q2.  Does that mean all the output explanations are accurate?
>
> For the seven foils we tested over five settings, the search found explanations that we had expected (they either detected concepts corresponding to a failing precondition or the cost function reported was defined using the expected concepts). As for more complex domains, we would argue that Montezuma's revenge is generally seen as a complex decision-making domain with a reasonably large state space. Since the main complexity of the search is sampling through the state space, we believe the performance of our system in Montezuma does reflect positively on its ability to scale our methods to larger domains.
>
> Q3.  it might be better to focus on steps where the foil has larger cost than proposed plan.
>
> We do mention in section 4, under cost function identification that we could just present a subset of actions to the user that is enough to establish the cost of the plan. We could identify such subsets by just performing a post-hoc processing of the full explanation generated by our method. We didn't use this during evaluation as we wanted to test the most basic version of the explanation without considering additional optimizations over it. Also, another discussion we didn't get to delve too much into was the other possible optimization schemes when searching for minimal abstractions. In this paper, we mostly focused on identifying the number of minimal concepts needed per step to refute the foil. An equally plausible optimization scheme may be to reduce the number of steps to be shown to the user.

---

> > ### Comment · AnonReviewer2 · 2020-11-23
> > **Response to rebuttal**
> >
> > Thanks for the response!
> >
> > About "Explanation confidence accuracy": Yes, my thought was simply that you can check if the explanations met your expectations. As you mentioned,  you have checked 7 foils under 5 settings, and all the highest-confidence explanations did meet your expectations, which is great. That partly justified that the confidence scores are accurate (at least for ranking).
> >
> > I'm still not a fan of the subjective setting of user study. However, except for that, I think my questions are well-addressed.

---

### Official Review · AnonReviewer3 · 2020-10-27
**Interesting setting for explanations of sequential decision making, but too many assumptions at play.**

**Rating:** 5
**Confidence:** 4

**Review:**


The authors propose a method of explainable AI for inscrutable blackbox models. The explanations build on a set of user-defined primitives, independently trained on the blackbox representation (e.g., visual frames of an Atari game), and use an increasingly popular method of providing contrastive explanations. Two forms of foil-based responses are provided: (1) indication of action failure from the planning perspective (preconditions unsatisfied); and (2) an explanation of relative sub-optimality that highlights key aspects of action costs that the user may be unaware of.

High-level concepts, particularly those tied to a symbolic description of the world dynamics, is an extremely compelling basis for explanation. It helps build a well-grounded intuition with human users/observers of autonomous systems and arguably is the best way to convey explanations that describe behaviour of an inherently sequential nature.

In addition to the form of explanation primitives, the algorithms are intuitive, and the probabilistic inference seems to be sound.

My concerns with the paper fall into two main categories: the lack of substantial contributions (particularly as related to representation learning) and the strong assumptions placed on the setting.

One of the most significant missed opportunities in this work is to focus on introducing new concepts. Especially given that human studies were conducted, and the setting was identified when algorithms fail, and new or revised concepts are required. Assuming highly accurate binary classifiers for each concept is relatively extreme, and it's only one such overly strong assumption.

Other very strong assumptions include:

(a) The state is memoryless/Markovian: every concept can be determined by looking exclusively at the current frame. This isn't the case in many settings where some memory of the previous actions is required.

(b) The distribution of a fluent across the state space is independent to the distribution of other fluents: this is rarely the case in planning-like domains, and the types of explanations introduced in this work build on planning-like domains a great deal.

(c) There is only one failed precondition: this might be an alright assumption to make given (b), but similarly, I find it unlikely that many domains would have this property.

As pointed out by the authors, assumptions (b) and (c) cause Algorithm 2 to exhaust the entire sampling budget before failing, and I don't believe they are safe assumptions to make.

On the topic of evaluation, there are two further issues. One is the scope of the evaluation (only two domains and a seemingly small number of subjects to test with), and this reduces the significance of the paper's contribution. Another issue is the choice of comparison for H3. The Saliency map is built using different information than that surfaced using the proposed approaches. This makes it challenging to adopt the experiment's conclusion as written since it is also testing the quality of the saliency map in addition to the comparative nature of the explanations.

I am leaning towards rejecting the paper due to the number of assumptions placed on the approach. Combined with the limited evaluation setting and scope of work, the contributions to the field of learning representations seem limited.

Ultimately, my hesitation in recommending acceptance comes from the contribution being on the low-side for the ICLR community. The authors identify key elements that would change this impression -- e.g., refining the concepts when the algorithms fail to find an explanation (italics on pg 5) -- but these do not play a central role in the proposed work.

The H1/H2 results are, in some sense, evident that the proposed explanations are preferred (19/20 for H1). While an important element (it would be very surprising if this weren't the case), they don't serve as a sufficient contribution in their own right. The H3 comparison seems to be somewhat contrived since they come from different sources -- a more accurate comparison would be to engineer the saliency overlay based on the domain-knowledge known. I.e., reflecting the precondition-based information directly. Without that, you conflate both the choice of focus and ability to highlight that choice.


Questions for the authors:

1. How do you remove the (seemingly strong) assumption that the distribution of fluents across the state space is independent among the fluents? Alternatively, why can we expect this to be a reasonable assumption to make?

2. How would you remove the dependence on the fully observable / Markovian assumption on the blackbox output that is used for concept classification? I.e., when the full state cannot be discerned by looking at the screen alone.


Other minor points of improvement for the paper:

o Mind the notation used for your Goal set (near the end of page 2, you are using a different syntax than the one introduced.

o Defn 3 seems to have a random bracket at the end.

---

> ### Author Response · Authors · 2020-11-11
> **Response to Review**
>
> Thank you for the comments, we have clarified some of the more important points below
>
> "(c) There is only one failed precondition: this might be an alright assumption to make given.."
>
> Please note that we do not make this assumption. There could be multiple failing preconditions at any given state, we just use one of the preconditions to refute the foil. That is rather than listing all failing preconditions, we list only a single one. This follows from the fact that failure of a single precondition is sufficient for the plan to fail and helps keep the explanation minimal.
>
> "Another issue is the choice of comparison for H3"
>
> The reason for choosing saliency maps as a baseline was both motivated by the popularity of saliency maps in explainable RL literature, and for this case the possibility of highlighting unmet preconditions. While we agree the methods don't use symmetric information, we would argue that the results show the need to invest in explanations that are explicitly in user vocabulary even for cases alternate methods could theoretically highlight similar information.
>
> "One of the most significant missed opportunities in this work is to focus on introducing new concepts."
>
> As detailed in our response to AnonReviewer 5, access to concepts is a common assumption made across many XAI works. Moreover, designing dialogues to actually acquire the concepts corresponds to an open HCI research problem and we feel the proposed explanation generation method on its own can add value to many scenarios.
>
> “My hesitation in recommending acceptance comes from the contribution being on the low-side for the ICLR community”
>
> About the relevance of our work to the ICLR community, we strongly believe that the interpretability of AI systems that learn their own representations will greatly benefit by the capability to provide post-hoc explanations in vocabulary that humans in the loop can understand.
>
> Questions:
>
> Q1. How do you remove the (seemingly strong) assumption that the distribution of fluents across the state space is independent among the fluents?
>
> Note that this is an assumption we make purely to simplify the confidence calculations, which is not the main contribution of the work. The probabilistic relationship between the concepts can easily be learned through interaction with the simulator (or from labeled traces) and once learned can be incorporated into the probabilistic graphical model. Incorporation of this knowledge only improves the accuracy of the confidence measure (and could be particularly useful when we have low accuracy classifiers since the presence of correlated concepts could be additional indication for the presence or absence of concepts). This change requires no fundamental modification in either search or inference procedures followed by the paper.
>
> Q2. How would you remove the dependence on the fully observable / Markovian assumption on the blackbox output that is used for concept classification?
>
> Note that our framework is capable of incorporating such features. The most straightforward way to incorporate such concepts would be to have users associate concepts to histories of observations. The symbolic model could very well be constructed over features that consider multiple low-level states. So even if the underlying system isn't Markovian the symbolic model could be. In terms of search, we will now need to sample over histories as opposed to individual states. In this work we focused on Markovian concepts since it is easier to analyze and run studies on, but we do see this work as the first step to supporting more general concept based explanations.

---

### Official Review · AnonReviewer1 · 2020-10-28
**Presents a novel explanation type and does so well**

**Rating:** 7
**Confidence:** 4

**Review:**

Edit:

I have read the other reviews as well as all author responses. The other reviewers noted meaningful concerns, but I believe the authors have clearly addressed most of these points. I still believe this work is an "accept".


Summary:

The authors present a system for producing explanations in terms of user-specified prerequisite relationships. The authors train a classifier to detect the presence of user-specified concepts. Relationships between these concepts are found by learning a partial symbolic model. With this model, an agent's action can be compared to a user-provided alternative and the first prerequisite-violating action can be identified (or the cost difference can be conveyed). The authors evaluate their approach on two domains with a user study.

Pros:

-This work addresses a relevant problem in explaining RL (and other SDM) agents and provides an initial solution that provides a solution for a meaningful set of domains.

-I agree with the authors: to my knowledge, this is the first work to provide explanations in terms of a learned symbolic model separate from the one used by the agent.

-The method for identifying failed preconditions is clearly introduced and motivated. The extension to handle noisy classifiers substantially increases the usefulness of this work.

Cons:

-This work requires a deterministic domain, as well as access to a simulator from which states can be sampled. This substantially limits the applicability of this approach. (This limitation is not mentioned in the abstract.)

-The authors make a number of assumptions (Section 4, "Confidence over explanations"), but these are not quantitatively evaluated. Adding experiments that measure how well these assumptions hold in practice would improve this work.

-The user study could be improved in a number of ways. It used manual translation of explanations to text. A comparison was made to saliency maps, but not to other explanations (such as causal explanations).

Questions During Rebuttal Period:

-Please address and clarify the "Cons" above.

-How important was the change to Montezuma's Revenge (rendering failed actions)? Is this a general requirement for creating explanations in an environment?

-Do you have additional information about the participants? Were they AI practitioners? Were any of the "main study" participants among those who selected the 25/38 concepts?

Other Comments:

-The authors may be interested in "Distal Explanations for Explainable Reinforcement Learning Agents" (a follow-up to the Madumal et al, 2020 work cited in Section 6).

-Section 5 appears to be broken into fewer subsections / has fewer line breaks in order to meet page requirements. This leads to a cramped, less organized Section 5.
Some Typos/etc:

-In general, another editing pass for articles and agreement of subject/verb plurality would help this paper.

-Line 2 of Introduction: "they ... with its" (change to "with their").

-Paragraph 3 of Introduction: The "'" after "questions of this form" is not matched.

-Background: "We will consider goal-directed agents that any given point is trying to drive" (fix verb agreement and remove extra words).

-The sentence leading into Definition 4 does not smoothly lead to Definition 4.

-Section 4: "So we start with" (remove "so").

-Section 4: "we aren't be able" (remove "be").

-Section 5: AdaBoost citation is using wrong command.

---

> ### Author Response · Authors · 2020-11-11
> **Response to Review**
>
> We thank the reviewers for all the comments and we will make sure to incorporate all fixes recommended by the reviewer into the paper.
>
> Q1 - Address the cons
>
> "work requires a deterministic domain and a simulator" -
> We chose to focus on the deterministic setting as we thought that would help us study the core of the explanation problem without being distracted by additional complexities posed by stochasticity. As specifically mentioned in Section 7, most of the methods discussed more or less directly apply to stochastic settings. The only major difference would be in how the foil is evaluated, i.e., instead of running a single simulation, we might need to test it multiple times and consider the expected value. Also in regards to simulators, please note that most RL systems in practice assume access to a simulator.
>
> "-The authors make a number of assumptions (Section 4, "Confidence over explanations"), but these are not quantitatively evaluated." -
> Most of the assumptions were satisfied by the Montezuma concepts by construction. For Sokoban, we expect to find correlated concepts since different users highlighted similar but distinct concepts for the same phenomena. As mentioned in our response to AnonReviewer3, the assumption about concept independence is something we can easily relax. As for the other assumptions, we will try to revise the paper with the analysis results on the degree to which the assumptions hold on sokoban before the end of the author rebuttal period.
>
> "The user study could be improved in a number of ways" -
> In terms of the explanation text, we have tried to make sure that the text could be something that could be easily generated through template filling and for sokoban names of the concepts are ones that participants provided. As for comparison to causal explanation, to the best of our knowledge that method requires access to the causal model upfront (at least the structure of it). This makes it incompatible to the central problem of reconciling vocabulary differences with the end user, since an end user would not be able to specify the causal structure of the domain.
>
>
> Q2 - How important was the change to Montezuma's Revenge?
>
> It was just added to create more diverse failure scenarios in the first level and isn't central to the method at all. The only requirement is for the simulator to be able to identify a failure state.
>
>
> Q3 - Do you have additional information about the participants?
>
> We have included demographics information in the Appendix. While we didn't ask them if they were AI practitioners, we did ask if they had taken any AI course and only half had taken AI courses for H1 and H2 (we tried to make sure we had participants from both groups when sending out the survey). While for H3, only 9 out of 24 participants reported ever taking an AI course.

---

> > ### Author Response · Authors · 2020-11-19
> > **Followup to rebuttal**
> >
> > We have included a comparison of empirically estimated target distributions with assumed distribution for both the second (in regards to the precondition) and third assumptions (in regards to the cost) in Appendix A.7

---

> > > ### Comment · AnonReviewer1 · 2020-11-24
> > > **Respose to Rebuttal**
> > >
> > > Thank you for the thorough answers to my questions and additional evaluation. My concerns have been sufficiently addressed.

---

### Official Review · AnonReviewer4 · 2020-10-28

**Rating:** 6
**Confidence:** 3

**Review:**

This paper presents a novel approach to generate contrastive explanations in a dialogue setting between a human and a planning agent. The setting assumes that the agent generates and offers an optimal plan to the user, and the user in turn challenges the presented plan offering an alternative (i.e. a contrast/foil). The goal of the agent is the denounce the alternative plan by explaining the infeasibility or suboptimality of the plan to the user in concepts they understand.

The explored direction is interesting and relevant as it seems to be a natural addition to the related problem of *generating* contrastive (a.k.a. counterfactual) explanations. I would suggest, however, that the paper more clearly distinguishes between the contrastive explanation generation literature (see, e.g., a survey [1]) with the type of explanation which is offered here, which is to identify the minimal set of preconditions (in an alternative concept space) that describes/explains the difference between two given instances (i.e., a model-proposed fact and a human-generated foil). This motivation is related to such (missing) related work as [2].


Strengths:
- the writing throughout well-polished and the motivation in the abstract and introduction is very well done
- the formulations are sensible and seem to be encapsulating the settings described and generalizations
- the inclusion of a user-study is helpful


Suggestions:
- the overloading of notation is at times difficult to follow
- as a reader not familiar with Montezuma's foils or Sokoban, I had a difficult time understanding the experimental section.
- a comparison with optimal explanations is lacking (relatedly, statements such as "the searchable to identify *the* expected explanation" is misleading, as, without an infinite budget and exhaustive search, the algorithm can identify an approximate to the optimal explanation)
- re: user study, I am not entirely convinced that the baselines are fair. by default, I would expect that offering more (non-fooling) information would render a higher subjective explainability score. perhaps the experiments would be stronger if tested against other types of information that is provided in addition to the presented baseline (something along the lines of H3; although even here it is not a completely fair comparison because unlike saliency maps, the offered explanations depend on human-annotated concepts which would naturally render the presented explanations are more human-understandable)
- nit; a completeness score of 3.36 / 5 is not possible (this would mean one of the 20 participants voted a non-integer value)
- in the related work, there seems to be an absence of literature on planning; perhaps a differentiation of the scope of the presented paper with this literature would boost the motivation


Summary:
I think this paper explores a fresh direction, which is necessary for enabling humans to contest, challenge, and ultimately trust the decisions of an automated system. I also believe that the presented material still requires a lot of further work and attention, and would be happy to see it accepted so the community can further explore these directions (fair user studies, the relation between approximate and optimal plans, investigations into the properties of foils relative to optimal actions, etc.). If accepted, I would strongly suggest a better integration of the main body and appendix, especially for Sec 3-5).


[1] Karimi et al., https://arxiv.org/abs/2010.04050
[2] Goyal et al., https://arxiv.org/abs/1904.07451

---

> ### Author Response · Authors · 2020-11-11
> **Response to Review**
>
> We thank the reviewer for the comments, below we have clarified some of the questions raised by the reviewer.
>
> "I am not entirely convinced that the baselines are fair"
>
> To the best of our knowledge, most of the symbolic explanation methods we were aware of either expected a full symbolic model to be specified in terms user can understand (c.f. Chakraborti et al. 2020) or expect information that can not be provided by the end-user (structural causal model of the task or positive and negative consequences for each action). Thus we argue that these simple baselines were in fact fair in so far they were the only ones that applied to the current problem.
>
> Additional References/Clarification about Planning related works:
>
> We thank the reviewer for the pointers to the papers, we will definitely make sure to include them. Also in terms of contrasting with planning works, as mentioned,  most of those works assume access to a symbolic model specified in user understandable terms. Will make sure to highlight this difference in the paper.
>
> Completeness Score:
>
> The completeness score was averaged across people who selected the explanation discussed in the paper (so in the case of H1 it was averaged across 19 participants). But double-checking the calculation we did find a small mistake in the average, the actual scores were 3.4746 for H1 and 3.2142 for H2 (as opposed to 3.35 and 3.36). We apologize for the mistake and will fix it in the paper.

---

> > ### Comment · AnonReviewer4 · 2020-11-24
> > **Response to rebuttal**
> >
> > I thank the authors for their response and for clarifying the relationship of this work to others in the field. I believe the community would benefit immensely if such a comparative discussion were amended to the manuscript and would improve the current exposition.

---

### Official Review · AnonReviewer5 · 2020-11-06
**Interesting problem, but unrealistic assumptions and unclear evaluations**

**Rating:** 5
**Confidence:** 4

**Review:**

This paper addresses the problem of answering queries about "foils,"
i.e., why an alternative plan was not chosen by an agent acting
optimally in a deterministic MDP. The authors describe three broad
classes of responses to this query: (1) one of the actions in the foil
does not satisfy the preconditions, (2) the foil does not achieve the
goal, or (3) the foil has suboptimal cost. Importantly, all responses
are conducted with respect to a pre-specified set of concepts
(predicates; or binary classifiers): the symbolic preconditions and
costs are learned by interaction with the simulator but expressed in
the language of these concepts. The authors discuss extensions to
their basic framework that (1) provide confidence measures along with
the responses and (2) handle noisy/probabilistic concepts.

A strength of this paper, in my opinion, is that it addresses a very
important problem and seems to make a lot of positive strides toward a
solution. I found the motivating paragraphs in the introduction to be
highly compelling, and the related work section to be sufficient for
placing this paper with respect to related literature on
explainability (which I am not very familiar with). Another plus point
is that the authors show that their system is robust to uncertainty
(via the confidence measures) and noise (via the probabilistic
concepts).

However, my main issue with this paper surrounds the assumption of the
pre-specified concepts. The most compelling motivational sentence to
me was, "More often than not...lay user"; however, if we are assuming
that lay users are the target users of the proposed system, where
would these concepts come from? I understand that they could be
specified ahead of time, but then there would be issues if the user
has a concept in mind while specifying a foil that is not covered by
the pre-specified set. The authors do partially address this concern
in the sentences "An empty list...task-related concepts.", but I did
not find this sentence compelling on its own, and it seemed to me that
the empirical studies do not consider this possibility of needing to
add more concepts to be able to answer a query (please do correct me
if I have missed this). To me, one of the most interesting aspects of
this problem is to consider how to automatically learn or improve the
initial set of concepts, perhaps given data of many users' foils
across different problem instances in a particular domain. For
instance, if we notice that users often provide a foil that involves
walking into the skull (we can figure this out with our simulator), we
may be able to learn that a concept NextToSkull is important when
building our symbolic preconditions. Unfortunately, the current paper
does not focus enough on this important aspect of the problem, instead
simply assuming a good set of concepts to be pre-specified, which I
find to be highly unrealistic.

A second major issue is the relatively low quality of the empirical
results. I appreciate that in this line of work, it can be hard to
provide rigorous numbers, since much of the evaluations come from
humans. Nevertheless, I believe that one can obtain much more
illuminating results in this problem setting by improving the
experimental design and reporting. A simple starting point would be to
include confidence measures on the reported numbers, in the paragraphs
discussing the results of H1/H2 and H3. Beyond that, it would be good
to probe deeper, e.g. for H3, cluster (and show us visualizations of)
the different situations where the precondition-based explanations
were useful while the saliency-based explanations were not, and vice
versa, so that we can better understand when each one tends to be
better. Personally, I could imagine many situations (e.g., in
understanding the behavior of an autonomous vehicle) where I would
rather just have the saliency map that highlights a region of my
surroundings, instead of being given a written explanation as to why
the vehicle performed a certain maneuver. So, I think the paper could
be greatly improved if the authors probe more deeply into explaining
the empirical results. By the way, much of the work by Anca Dragan (I
am unaffiliated), see e.g. [1], can be a useful reference in how to set up highly
rigorous user studies.

Looking forward, I have one question for the authors. It is clear that
in real-world applications of AI, our agents will never be able to act
optimally, and we shouldn't expect them to. Given this, would there be
a way to modify the current work to create feedback from the user that
allows the agent to improve its solution? For instance, perhaps the
agent is in the middle of executing the best solution it found after
thinking for 30 minutes, but then the user says "why not do X
instead?", and the agent has to decide between: (a) *the user* missed
something, and I should generate an explanation; or (b) *I* missed
something, and I should think more and revise my current policy.

[1] Bobu, Andreea, et al. "Learning under Misspecified Objective Spaces." Conference on Robot Learning. 2018.

---

> ### Author Response · Authors · 2020-11-11
> **Response to Review**
>
> We thank the reviewer for their comments and suggestions, here are some clarifications on some of the questions raised
>
> “where would these concepts come from?”
>
> We start by noting that assuming availability of concepts as an initial step is a common assumption in explainable AI (for example TCAV systems - Kim et al. 18). In practice, the realizations of these methods usually rely on pre-collecting a database of concepts that would be meaningful to the users. This approach is particularly useful when you have access to users with similar backgrounds and they have some experience with the task/system. Such concepts could also be extracted from possible textual descriptions of the domain/task.
>
> Allowing for concept acquisition:
> We agree that this assumption may not always hold and unlike these previous works our method specifically allows for the possibility that the concept list may be incomplete. The reason we haven’t tested such scenarios yet, is that setting up the dialogue with the users to acquire these new concepts requires solving a non-trivial HCI problem.
> Even though the general flow of such an interaction is more or less clear (eg: We could ask users for the concepts by contrasting the failing state and a number of states where the action is successful). There are a number of open questions like:
> Q1) what strategy should we use to select the positive states? should we look at states that are visually similar to the current failing state or distinct
> Q2) what supplementary information could we include that could help the user identify missing concepts? for example listing already rejected concepts or even using saliency maps to highlight regions.
> We believe the design of such interaction schemes to be a separate research question and the proposed explanation method still adds value on its own.
>
>
> "I could imagine many situations (e.g., in understanding the behavior of an autonomous vehicle) where I would rather just have the saliency map that highlights a region of my surroundings"
>
> The specific hypothesis we wanted to test was whether the concept based explanation helps users better understand the task. Our own intuition for testing this hypothesis was based on the fact that even the best saliency map could highlight task-irrelevant artifacts and the users may end up misattributing concepts that correspond to the highlighted region. This is in addition to cases where saliency maps may not be enough to capture the underlying information, for example when the features may be non-markovian (please refer to our discussion with AnonReviewer3 on Q2). Similar drawbacks of saliency maps have been also reported by works like (Kim et al. 2018) and (Atrey et al. 2019). Though we agree with the reviewer that in cases where a quick response from the user is crucial and the user is familiar enough with interpreting saliency maps and there is no ambiguity, such highlighting could act as a shorthand and help users make quick decisions. We will update our hypothesis and results to clarify that we are not considering such scenarios.
>
> " I missed something, and I should think more and revise my current policy."
>
> We agree that foils of the type studied in the paper make for an excellent source of feedback about not only information about possible suboptimality of the policy (which could be tested via the simulator), but also about the task and even latent user preferences. Once the system gives its explanations, the user can use that information to either update their own understanding of the task or if they are confident about their understanding they can use the generated explanatory messages as a starting point to help improve the system performance. Such interactions are completely complementary to the explanation system described in the paper, and there certainly are complementary works in planning literature which have looked at supporting such interactions (C.f Valmeekam et al. 2020), it would be interesting to see how we could incorporate such interaction in our more general settings.
>
>
> We also thank the reviewer for their comments on including the additional measures for the reported statistics. We will make sure to revise the paper with the new measures soon.
>
>
> Kim, B., Wattenberg, M., Gilmer, J., Cai, C., Wexler, J., & Viegas, F. (2018, July). Interpretability beyond feature attribution: Quantitative testing with concept activation vectors (tcav). In International conference on machine learning (pp. 2668-2677). PMLR.
>
> Atrey, Akanksha, Kaleigh Clary, and David Jensen. "Exploratory Not Explanatory: Counterfactual Analysis of Saliency Maps for Deep Reinforcement Learning." International Conference on Learning Representations. 2019.
>
> Valmeekam, Karthik, Sreedharan, Sarath, Sengupta, Sailik and Kambhampati, Subbarao “RADAR-X: An Interactive Interface Pairing Contrastive Explanations with Revised Plan Suggestions.” ICAPS Workshop on Explainable AI Planning (XAIP), 2020

---

> > ### Author Response · Authors · 2020-11-19
> > **Followup to rebuttal**
> >
> > We have included the 95% confidence intervals for the results corresponding to H3 in appendix A.6

---

> > ### Comment · AnonReviewer5 · 2020-11-22
> > **Response to rebuttal**
> >
> > Thank you to the authors for the detailed responses and improvements to the reporting of results. In my opinion, concept acquisition is such a fundamental and necessary aspect of the proposed system that I believe it needs to be evaluated in order for us to truly understand the practical benefits of the system. As of now, I will increase my score (but still slightly lean reject); after discussion with the other reviewers, I may be willing to switch to accept if the general consensus among them is that the impact of the approach is sufficient as-is.

---

### Author Response · Authors · 2020-11-19
**Change Logs**

We thank the reviewers for all the comments and suggestions, we have updated the current draft of the paper to reflect some of these comments

[AnonReviewer 1] Have included empirical evaluation of the assumptions on Sokoban variants in appendix A.7

[AnonReviewer 4] Have fixed the completeness score

[AnonReviewer 5] Have included the confidence intervals for results reported for H3 under appendix A.6

[AnonReviewer 1/AnonReviewer 2/AnonReviewer 3] Incorporated the typos pointed out by the reviewers

---

### Decision · Program_Chairs · 2021-01-07
**Final Decision**

**Decision:**

Reject

**Comment:**

This paper is an intriguing study of agents that can give explanations (contrastive) of their actions via symbolic representation such as dialog.  Agents can also allow users to argue against the agents' decisions. I am extremely impressed by the quality of the reviewer comments and discussions.  It is also interesting that the reviewers have formed two camps of thought on the paper: One camp consists of R3 and R5 who are in agreement in vociferously criticizing the weak points in the paper.  The other camp consists of R1, R2, and R4, who champion the merits of what they see as strong points.  Notably, all reviewers have fairly high confidence values -- only one confidence score of 3 and all others are 4.

It was a borderline case and not an easy decision. In the end the program committee decided that the paper in its current form does not quite meet the bar, and would benefit from another revision (see e.g., R4 comments).  We think that the work is interesting, and encourage the authors to address the reviewers' comments and resubmit the work to another venue.